# Development of an improved blood-stage malaria vaccine targeting the essential RH5-CyRPA-RIPR invasion complex

Barnabas G. Williams [1,2,3,11], Lloyd D. W. King [1,2,3,11], David Pulido[3], Doris Quinkert [1,2,3], Amelia M. Lias [1,2,3], Sarah E. Silk[1,2,3], Robert J. Ragotte [1,3], Hannah Davies [1,2,3], Jordan R. Barrett[1,2,3], Kirsty McHugh[1,2,3], Cassandra A. Rigby[1,2], Daniel G. W. Alanine[1,3], Lea Barfod [3], Michael W. Shea[3], Li An Cowley [1,3], Rebecca A. Dabbs[3], David J. Pattinson[3], Alexander D. Douglas [3], Oliver R. Lyth[1,3], Joseph J. Illingworth[3], Jing Jin[3], Cecilia Carnrot [4], Vinayaka Kotraiah[5], Jayne M. Christen[5], Amy R. Noe [5,10], Randall S. MacGill [6], C. Richter King[6], Ashley J. Birkett[6], Lorraine A. Soisson[7], Katherine Skinner[1,2,3], Kazutoyo Miura [8], Carole A. Long [8], Matthew K. Higgins [1,2] & Simon J. Draper [1,2,3,9] ✉

Reticulocyte-binding protein homologue 5 (RH5), a leading blood-stage *Plasmodium falciparum* malaria vaccine target, interacts with cysteine-rich protective antigen (CyRPA) and RH5-interacting protein (RIPR) to form an essential heterotrimeric "RCR-complex". We investigate whether RCR-complex vaccination can improve upon RH5 alone. Using monoclonal antibodies (mAbs) we show that parasite growth-inhibitory epitopes on each antigen are surface-exposed on the RCR-complex and that mAb pairs targeting different antigens can function additively or synergistically. However, immunisation of female rats with the RCR-complex fails to outperform RH5 alone due to immuno-dominance of RIPR coupled with inferior potency of anti-RIPR polyclonal IgG. We identify that all growth-inhibitory antibody epitopes of RIPR cluster within the C-terminal EGF-like domains and that a fusion of these domains to CyRPA, called "R78C", combined with RH5, improves the level of in vitro parasite growth inhibition compared to RH5 alone. These preclinical data justify the advance-ment of the RH5.1 + R78C/Matrix-M™ vaccine candidate to Phase 1 clinical trial.

The deadliest form of human malaria is caused by the apicomplexan parasite *Plasmodium falciparum*; transmitted by the bite of the female *Anopheles* mosquito. Malaria deaths declined steadily for more than a decade but recently increased to 608,000 in 2022 with 55,000 additional deaths linked to the COVID-19 pandemic[1]. Therefore, the development of safe, effective, and durable malaria vaccines remains a global public health priority[2]. Two malaria vaccines, RTS,S/AS01, and R21/Matrix-M™, have now received World Health Organisation (WHO)

[1]Department of Biochemistry, University of Oxford, Dorothy Crowfoot Hodgkin Building, Oxford, UK. [2]Kavli Institute for Nanoscience Discovery, Dorothy Crowfoot Hodgkin Building, University of Oxford, Oxford, UK. [3]The Jenner Institute, University of Oxford, Old Road Campus Research Building, Oxford, UK. [4]Novavax AB, Kungsgatan 109, SE-753 18, Uppsala, Sweden. [5]Leidos Life Sciences, Frederick, MD, USA. [6]Center for Vaccine Innovation and Access, PATH, Washington, DC, USA. [7]USAID, 1300 Pennsylvania Ave. NW, Washington, DC, USA. [8]Laboratory of Malaria and Vector Research, NIAID/NIH, Rockville, MD, USA. [9]NIHR Oxford Biomedical Research Centre, Oxford, UK. [10]Present address: Latham BioPharm Group, Elkridge, MD, USA. [11]These authors contributed equally: Barnabas G. Williams, Lloyd D. W. King. ✉e-mail: simon.draper@bioch.ox.ac.uk

prequalification for use in young children[3]. Both are similar in design, targeting the circumsporozoite protein (CSP) on the pre-erythrocytic sporozoite stage of the parasite and inducing antibodies that prevent infection of the liver. However, when a single sporozoite slips through this protective net a productive infection is initiated and, following liver-stage development, merozoites emerge into the blood where they undergo exponential growth leading to clinical disease. Indeed, the development of a vaccine that can effectively block merozoite invasion into host red blood cells (RBC) may provide a second layer of protection against clinical disease, death, and onward transmission when combined with the existing vaccines that target CSP in a multi-stage approach[2].

Merozoites invade RBCs through a complex interplay of host-parasite receptor-ligand interactions. Redundancy of these invasion pathways and substantial strain-to-strain variation of other blood-stage antigen targets[4,5] hindered blood-stage vaccine development efforts for many years. The discovery that *P. falciparum* reticulocyte-binding protein homologue 5 (RH5) is highly conserved, forms an essential interaction with basigin (BSG/CD147) on the human erythrocyte[6–9], and is susceptible to vaccine-induced broadly neutralising antibodies[10,11] has led to a renewed vigour in this field of research. Clinical trials of the first vaccine candidates targeting the full-length RH5 molecule have since demonstrated the induction of cross-strain growth-inhibitory antibodies[12] and significantly reduced the growth rate of *P. falciparum* in the blood of healthy adults following vaccination and controlled human malaria infection[13]. Moreover, highly promising RH5 vaccine candidate immunogenicity in African infants, a critical target population for *P. falciparum* malaria vaccines, has since been reported[14]. Here, levels of in vitro growth inhibition activity (GIA) achieved using purified total IgG against *P. falciparum* blood-stage parasites greatly exceeded those observed in adult vaccinees from non-endemic countries; moreover, these levels of GIA in vaccinated infants were now reaching levels previously defined as protective[15], and mechanistically correlated[16], in non-human primates. The current leading vaccine candidate, soluble recombinant protein RH5.1[17] formulated with Matrix-M™ adjuvant, has since entered Phase 2b field efficacy testing in West Africa (ClinicalTrials.gov NCT04318002 and NCT05790889).

RH5 is delivered to the apical surface of *P. falciparum* merozoites along with cysteine-rich protective antigen (CyRPA)[18] and RH5-interacting protein (RIPR)[19], with which it forms an essential hetero-trimeric complex (RCR-complex)[8,20]. Like RH5, the components of the RCR-complex appear to be poor targets of naturally-acquired malaria immunity and thus highly conserved[20]. Structurally RH5 forms a diamond-like architecture composed of two three-helical bundles, with BSG binding across the tip of RH5[6]. CyRPA forms a 6-bladed β-propeller (6BBP) structure[21,22] that bridges the base of the RH5 helical diamond and the N-terminal core domain of RIPR[23,24]. Most recently, two further protein components have been shown to bind the RCR-complex, *Plasmodium* thrombospondin-related apical merozoite protein (PTRAMP) and small cysteine-rich secreted protein (CSS). These form a disulphide-linked heterodimer which bridges from the merozoite surface to the C-terminal tail of RIPR thereby forming a penta-meric complex (PCRCR)[24,25]. The conserved and essential nature of these targets has now raised the prospect of defining new and improved blood-stage vaccine candidates that target this wider invasion complex as opposed to RH5 alone.

Encouragingly, as for RH5, studies in various animal models have consistently shown that vaccination with the full-length CyRPA[18,26–33] and RIPR[19,30,34] antigens can induce functional growth-inhibitory polyclonal antibodies. This work has been extended through the study of monoclonal antibodies (mAbs) that have identified antibody-susceptible epitope regions of these molecules including the top of the RH5 helical diamond close to or overlapping with the BSG-binding site[6,35,36], as well as blades 1 and 2 of the CyRPA 6BBP[22,37]. In contrast,

detailed information regarding the location of potent epitopes for RIPR is lacking. In addition, vaccine-induced polyclonal antibodies and/or mAbs against RH5, CyRPA and/or RIPR have been reported to show additive or synergistic functional growth inhibition against *P. falciparum* using in vitro assays or in vivo using humanised mouse challenge models[18,21,27,28,30,31,36–38]; however, this has not been systematically analysed. Nevertheless, these data suggest a multi-antigen vaccine candidate strategy could achieve significantly higher efficacy and/or durability via induction of a more potent growth-inhibitory antibody response. Here, we therefore sought to investigate whether a vaccine candidate based on the ternary RCR-complex could improve upon the leading clinical candidate vaccine RH5.1/Matrix-M™.

## Results

### Reconstitution of the RCR-complex in vitro

To study the function of the RCR-complex, we produced all three full-length antigens as soluble recombinant proteins, each with a C-terminal four amino acid C-tag for purification[39]. RH5 and RIPR were expressed using a *Drosophila* S2 cell platform[40], while CyRPA was expressed from mammalian HEK293 cells[41]. We sought to reconstitute in vitro all possible binary complexes as well as the ternary complex by incubating these proteins in equimolar ratios. We were only able to reconstitute the binary combinations of RH5+CyRPA and RIPR+CyRPA, as well as the ternary RCR-complex, as analysed and purified by size exclusion chromatography (SEC). SEC peaks, corresponding to intact complexes, dissociated into individual proteins under non-reducing SDS-PAGE conditions (Fig. 1A–C). We were unable to reconstitute the binary complex between RH5 and RIPR, in line with CyRPA acting as the central adaptor molecule within the ternary RCR-complex[23,24,42]. Having reconstituted the ternary RCR-complex we next sought to characterise its interactions with a panel of mAbs against all three target antigens (Table S1).

### All in vitro growth inhibitory mAbs bind the ternary RCR-complex

We initially analysed a panel of twenty RH5-specific mAbs which, as we have previously shown, can be used to identify seven distinct epitope patches around the RH5 molecule[6,36] (Table S1). The brown, blue, and red epitope patches are clustered around the diamond tip of RH5, overlapping with or very close to the basigin binding site[35,36]. The orange, yellow, and purple epitope patches are located around the bottom of the RH5 diamond, overlapping with or very close to the RH5-CyRPA interaction site[23,24,36,37]. The final green epitope patch, which includes non-growth inhibitory antibodies that can synergise with other growth inhibitory antibodies, is found within a central region of the RH5 diamond structure, close to the site where the disordered N-terminal region joins the structured region of the RH5 molecule[36].

We previously reported the ability of each anti-RH5 mAb to mediate in vitro GIA against 3D7 clone *P. falciparum* parasites[36]. Strong GIA is observed for mAbs binding epitope patches close to, or overlapping with, the basigin binding site, however no GIA was detected for the mAbs that bound the other epitope patches (Supplementary Fig. 1A). To extend this work, we now explored the relationship between epitope accessibility and the GIA exhibited by antibodies that bind to all three antigens within the RCR-complex. Here we expanded our panel of mAbs to include a set of anti-CyRPA mAbs[37,43], and a set of novel mouse derived anti-RIPR mAbs that displayed a spectrum of GIA (Table S1, Supplementary Fig. 1B, C).

We first incubated each mAb with the pre-formed RCR-complex before analysis by SEC. These data allowed us to identify and define two mAb "types": Type I antibodies were those that could form a stable quaternary complex, seen as a shift to a larger elution volume by SEC, when incubated with the pre-formed RCR-complex. Type II antibodies were unable to bind the pre-formed RCR-complex, seen as two distinct peaks on SEC; the first for the RCR-complex alone and a second for the

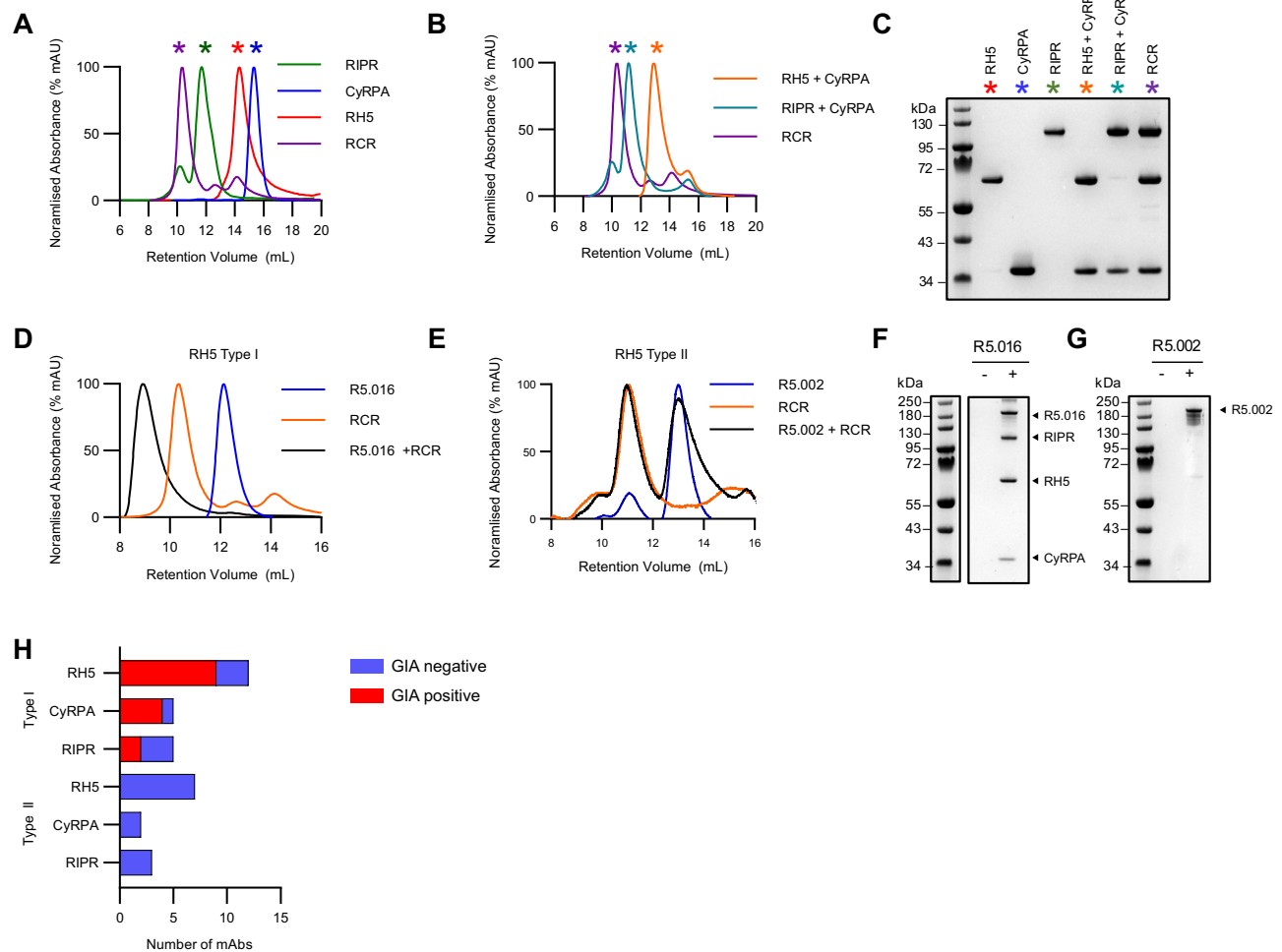

**Fig. 1 | Characterisation of mAb binding to the RCR-complex. A** Size exclusion chromatograms demonstrating RCR-complex formation between RH5, CyRPA and RIPR, and (**B**) demonstrating binary complex formation between RH5+CyRPA, and RIPR+CyRPA. **C** Non-reducing SDS-PAGE gel assessing binary and ternary complex formation. Coloured asterisks on chromatograms indicate which gel lanes correspond to the peaks in panels (**A**) and (**B**). Representative example of three independent experiments shown. **D** Size exclusion chromatogram showing a representative example of complex formation analysis between the RCR-complex and anti-RH5 mAb R5.016 [Type I, GIA-positive], and (**E**) anti-RH5 mAb R5.002 [Type II, GIA-negative]. **F** Non-reducing SDS-PAGE gel of co-immunoprecipitation of the pre-formed recombinant RCR-complex without (-) or with (+) mAb R5.016 [Type I, GIA-positive] or (**G**) mAb R5.002 [Type II, GIA-negative] bound to protein G agarose beads. Representative examples are shown. A representative example of two independent experiments is shown. **H** Bar chart summarising the number of Type I and Type II mAbs for each antigen and whether these mAbs show GIA. Source data are provided as a Source Data file.

unbound mAb (Fig. 1D, E and Supplementary Fig. 1D–H). We also confirmed these results by pull-down immunoprecipitation (Fig. 1F, G and Supplementary Fig. 1D–H). Here, mAbs were incubated with the preformed RCR-complex; when a Type I mAb was used, both the mAb and RCR-complex could be recovered using Protein G beads. However, when a Type II mAb was used, only the mAb was recovered using Protein G beads. This analysis thus identified that all GIA-positive mAbs, regardless of the target antigen, are Type I, i.e., their epitope is exposed in the RCR-complex and they can bind to form a quaternary complex. In contrast, GIA-negative mAbs can be either Type I or Type II (Fig. 1H and Table S1). This conclusion is also supported by structural data that are available for a subset of the anti-RH5[6,36] and anti-CyRPA[24,37] mAbs. Epitopes for mAbs with known anti-parasitic growth-inhibitory properties are exposed on the formed RCR-complex, whilst the epitopes of mAbs that do not inhibit parasite growth are often masked (Fig. 2). No structural data are currently available for anti-RIPR mAb complexes, however, it was notable this panel of mAbs displayed the highest proportion (60%) of GIA-negative Type I antibodies, suggesting many exposed epitopes on RIPR within the RCR-complex do not induce neutralising antibodies.

## Pairs of anti-RH5, -CyRPA, and -RIPR antibodies show inter-antigen synergistic GIA

Having assessed individual mAbs for GIA, and having shown these epitopes are exposed within the formed RCR-complex, we next sought to define the efficacy of different antibody combinations. We have previously reported that specific combinations of anti-RH5[36] or anti-CyRPA[37] mAb clones show "intra-antigen" synergistic GIA. Here, we undertook a comprehensive GIA analysis to evaluate "inter-antigen" antibody interactions across the RCR-complex. We selected four anti-RH5 mAbs (R5.004, R5.008, R5.011 and R5.016), three anti-CyRPA mAbs (Cy.003, Cy.007 and Cy.009), and one anti-RIPR mAb (RP.012), as representative clones from the non-overlapping Type I epitope sites on each antigen and tested these in pair-wise combinations for synergistic GIA using the Bliss definition of additivity[30,44]. Weak synergistic interactions were observed for the majority of mAb combinations (Fig. 3A and Supplementary Fig. 2), however clear 'hotspots' of synergy were identified with combinations including the anti-RH5 mAbs R5.008 and R5.011 (Fig. 3B). We have previously reported the ability of the GIA-negative R5.011 mAb to synergise with or "potentiate" anti-RH5 growth inhibitory antibodies[36]. Here, we

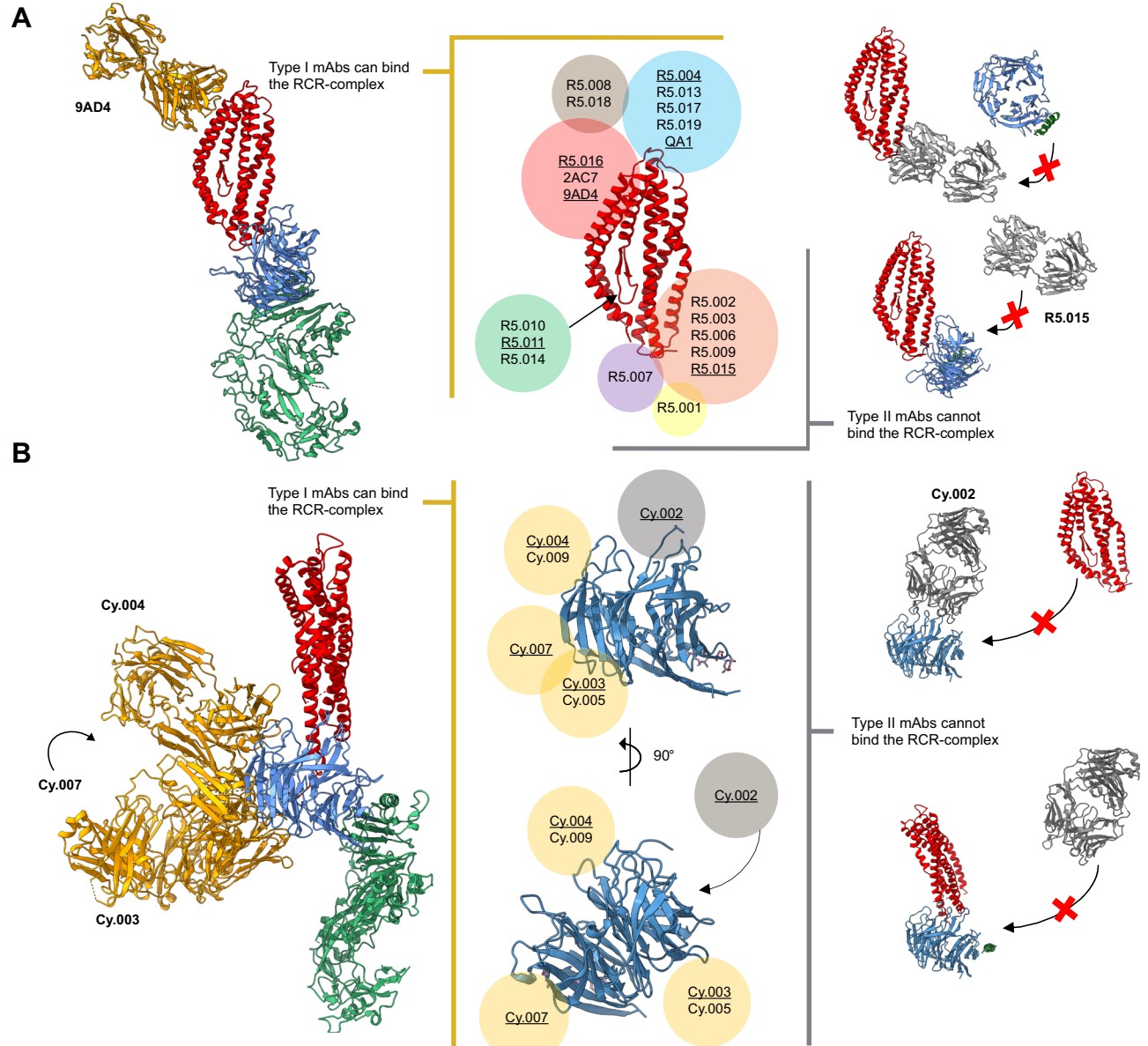

**Fig. 2 | Available structural data concur with classification of mAbs as Type I or Type II. A** Crystal structure of RH5 (centre, red) with mAb epitope bins overlaid (coloured circles). Antibody clusters were identified as Type I (left, yellow box) and Type II (right, grey box) depending on their ability to bind to the RCR complex. Anti-RH5 mAbs with available crystal structures of their Fab bound to RH5 are underlined. The RH5 (red), CyRPA (blue), RIPR (green) and 9AD4 or R5.015 Fab complex structures (left and right of centre, respectively) are a composite of published structures (PDB: 4U0Q[6], 7PHU[37], 6MPV[23] and 8CDD[24]) that concur with Type I (gold) or Type II (grey) mAb classification. **B** Crystal structure of CyRPA (centre, blue) with mAb epitope bins overlaid (coloured circles). Antibody clusters were identified as Type I (left, gold circles) and Type II (right, grey circles) depending on their ability to bind to the RCR complex. Anti-CyRPA mAbs with available crystal structures of their Fab bound to CyRPA are underlined. The RH5 (red), CyRPA (blue), RIPR (green) and anti-CyRPA Fab (Cy.002, Cy.003, Cy.004, Cy.007) complex structures (left and right of centre, respectively) are a composite of published structures (PDB: 7PI3[37], 7P17[37] and 6MPV[23]) that concur with Type I (gold) or Type II (grey) mAb classifications. The full RIPR structure has been excluded from the Type II mAb illustrations for clarity.

identified that the GIA-positive clone, R5.008, could also synergise. When this mAb was held at a constant concentration, the anti-RIPR mAb and all the anti-CyRPA mAbs tested in combination showed synergy with a range of 2- to 5-fold improvement in GIA over the predicted additivity under the test conditions. This was particularly pronounced with the weakly GIA-positive mAb Cy.003 (Fig. 3B, C). These data indicate that at least two anti-RH5 epitope specificities have the potential to synergise with anti-CyRPA and anti-RIPR antibody responses, whilst the others combine at least additively. Critically, no RH5 / CyRPA / RIPR mAb combination was shown to be antagonistic, further supporting the rational for RCR-complex-based vaccine strategies.

## Immunisation with the RCR complex leads to immune competition and a suboptimal response
The above data suggested all growth-inhibitory epitopes are exposed on the formed RCR-complex, and that antibody responses across the three antigens could act additively, if not synergistically. We, therefore, hypothesised that in vivo immunisation with the formed RCR-complex could induce a polyclonal antibody response that improves upon the use of single antigen vaccines. To investigate this, we immunised cohorts of Wistar rats three times with full-length RH5, CyRPA, RIPR, pairwise combinations, a mix of all three antigens, or the pre-formed RCR-complex; all vaccines were formulated in Matrix-M™ adjuvant. Three approaches were used for the triple antigen "RCR"

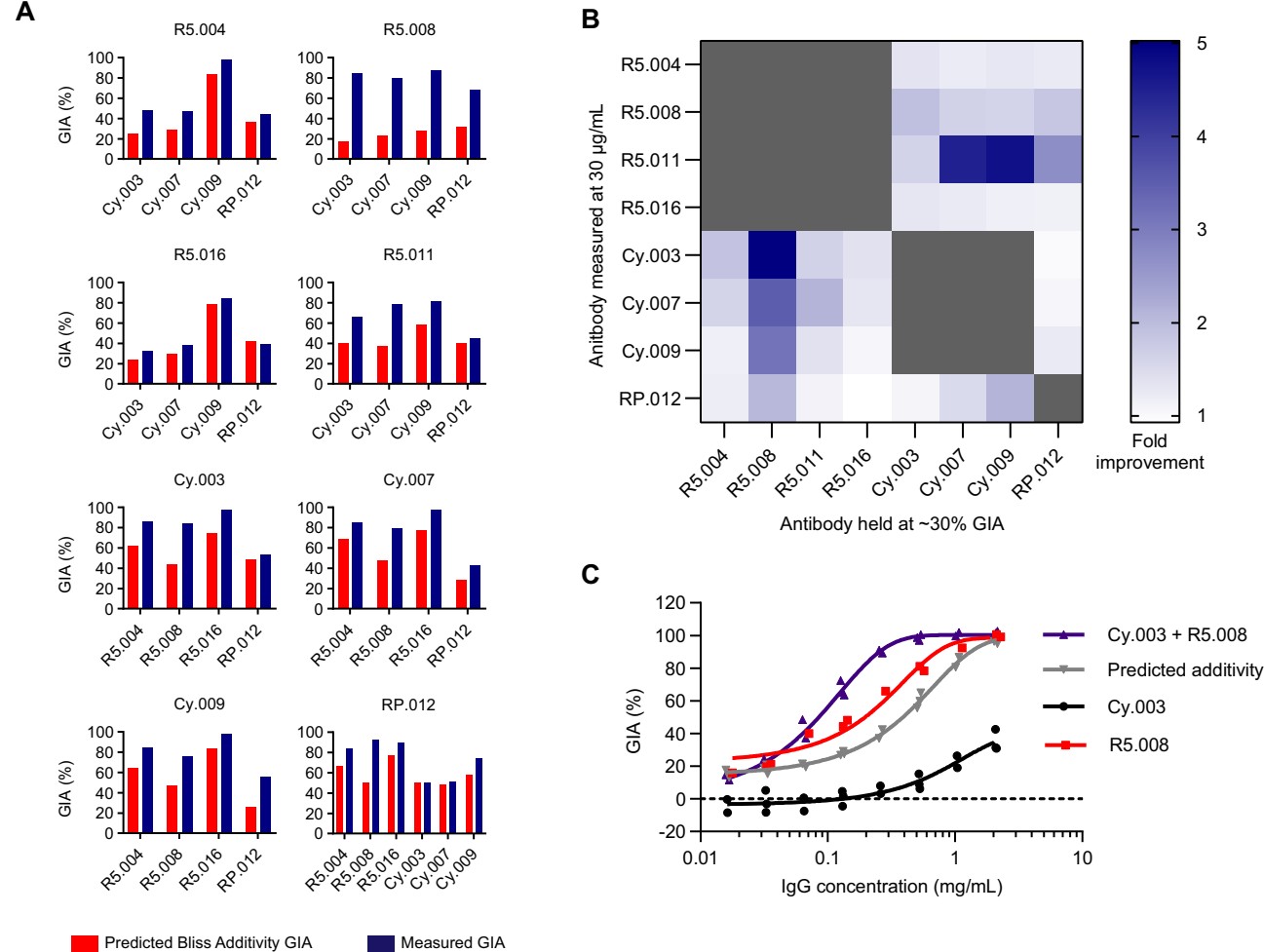

**Fig. 3 | Demonstration of synergistic inter-antigen GIA of mAbs targeting the RCR-complex. A** Predicted growth inhibitory activity (GIA) based on Bliss additivity (red) compared to measured GIA (dark blue) for a mAb combination where one was held at 30 % GIA (X-axis) and the other measured at ~300 µg/mL (Title); R5.011 was held at 2 mg/mL (not 30 % GIA) due to R5.011 alone having no GIA activity. Complete dilution curves are shown in Figure S2. The bar indicates the mean across triplicate measurements. **B** Heat map summary of the fold improvement over Bliss additivity across the mAb panel. **C** Synergy GIA analysis of anti-RH5

mAb R5.008 combined with anti-CyRPA mAb Cy.003 in a 1:1 mixture (i.e., 1 mg/ mL = 1 mg/mL Cy.003 or 0.5 mg/mL Cy.003 + 0.5 mg/mL R5.008). Grey: Cy.003 alone titration curve. Black: R5.008 alone titration curve. Red: predicted Bliss additivity GIA for a 1:1 mixture of Cy.003 and R5.008. Blue: measured data. Each data set fitted with a Richard's five-parameter dose-response curve with no constraints. Individual data points are the mean of triplicate wells in each experiment; $N = 2$ independent experiments for R5.008 and $N = 3$ for Cy.003 alone and Cy.003 + R5.008 (1:1 mixture). Source data are provided as a Source Data file.

immunisations: for two groups, the antigens were admixed in adjuvant at the point of administration, and protein doses were either matched to the single antigen vaccines ("R + C+Ri" group, i.e., 2 µg RH5 + 20 µg CyRPA + 20 µg RIPR), or equimolar with the RCR-complex immunised group ("R + C+Ri Equimolar" group, i.e., 5.3 µg RH5 + 3.5 µg CyRPA + 11.1 µg RIPR); for the third group, RH5, CyRPA, and RIPR were mixed and purified as the RCR-complex before mixing with adjuvant and then immunisation.

We observed some increases in anti-RH5 responses after the third vaccine dose in rats, especially in the mixed antigen groups. However, anti-RH5 IgG responses remained significantly lower in mixed antigen groups as compared to immunisation with RH5 alone; apart from the RH5+CyRPA group (Fig. 4A, Supplementary Fig. 3A and Table S2A). Similarly, anti-CyRPA IgG responses were significantly reduced by between 2- and 10-fold after the third dose when co-immunising with a second or third antigen (Fig. 4B, Supplementary Fig. 3A and Table S2B). In contrast, we observed that anti-RIPR IgG responses peaked at >1000 µg/mL after the first boost and did not increase further; moreover, these concentrations remained relatively high for all groups, even when co-immunising with CyRPA and RH5

(Fig. 4C, Supplementary Fig. 3A and Table S2C). Neither anti-RH5 nor anti-CyRPA responses approached the quantitative magnitude of the anti-RIPR IgG response. These data suggest that full-length RIPR is immuno-dominant and can suppress the antibody response against RH5 and CyRPA following co-immunisation.

Purified total IgG (measured in mg/mL) was subsequently tested for in vitro GIA (Supplementary Fig. 3B). To link the GIA observed with total purified IgG to the total vaccine-induced antibody response, the IgG responses to each antigen in each purified total IgG sample (as measured by standardised ELISA in arbitrary units [AU]) were converted to µg/mL by calibration-free concentration analysis (CFCA)[12] (Supplementary Fig. 4A) and then summed. We further validated this approach by developing an "RH5+CyRPA+RIPR" standardised ELISA which independently measured the total combined antigen-specific response, in AU, in a single assay. The reported "RH5+CyRPA+RIPR" standardised ELISA AU, and the summed µg/mL derived from individual standardised ELISAs converted by CFCA, significantly correlated (Supplementary Fig. 4B, C), suggesting both assay formats give similar results. We therefore elected to use the summed µg/mL method for reporting the total vaccine-induced

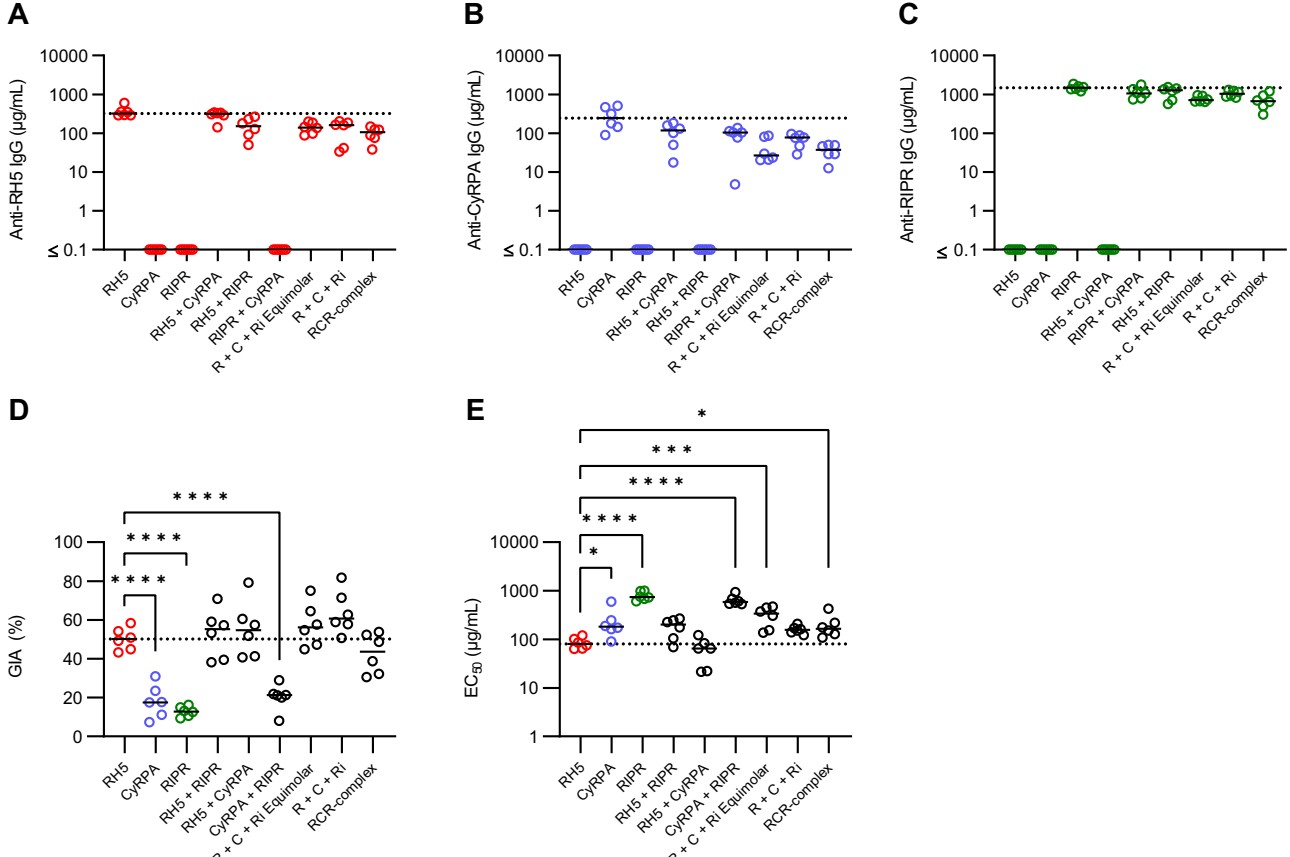

**Fig. 4 | Immunisation with combinations of RH5, CyRPA and/or RIPR does not improve over immunisation with RH5 alone.** Wistar rats were immunised on days 0, 28 and 56. Terminal bleed was taken on day 70 (post-third dose). Doses of 2 µg RH5, and 20 µg CyRPA and RIPR antigen were used. For the "R + C+Ri" group, antigen doses were matched to the single antigen vaccination groups (2 µg RH5 + 20 µg CyRPA + 20 µg RIPR). For the "R + C+Ri Equimolar" group antigens were dose-matched to the RCR-complex (5.3 µg RH5 + 3.5 µg CyRPA + 11.1 µg). Day 70 serum IgG ELISA data (reported in µg/mL) shown against full-length (**A**) RH5 (red), (**B**) CyRPA (blue), and (**C**) RIPR (green). The dotted line corresponds to the median antigen-specific IgG response from the relevant group of single antigen-immunised animals. A summary of statistical analysis related to these panels can be found in Table S2. **D** Single-cycle GIA assays were performed using *P. falciparum* clone 3D7. Total IgG, purified from day 70 serum samples, was titrated in the GIA assay (see

Supplementary Fig. 3B). GIA at 1 mg/mL total purified IgG was interpolated for each animal. The dotted line indicates the median RH5 GIA. Significance determined by one-way ANOVA with Dunnett's multiple comparisons test versus RH5 group only, **** $p < 0.0001$. **E** Data from (**D**) were replotted against total antigen-specific IgG concentration in µg/mL as measured by ELISA in each purified total IgG sample (see Supplementary Fig. 3C). Each dataset was fitted with a Richard's five-parameter dose-response curve with no constraints to ascertain the GIA assay $EC_{50}$. The dotted line shows the median result for the RH5 only group for comparison. Individual and median group responses ($N = 6$ per group) are shown in all panels. Significance was determined by one-way ANOVA of log-transformed data with Dunnett's multiple comparisons test versus the RH5 group only. * $p < 0.05$, ** $p < 0.001$, *** $p < 0.0001$, **** $p < 0.0001$. Source data are provided as a Source Data file.

response since this enabled quantitative comparison between vaccine groups. We next replotted the GIA data versus total antigen-specific IgG measured by ELISA and converted to µg/mL by CFCA to assess the overall functional quality of the vaccine-induced antibodies (Supplementary Fig. 3C), and from these data interpolated the GIA $EC_{50}$, i.e., the concentration of total antigen-specific IgG needed to achieve 50 % GIA. No antigen combination showed significantly improved overall GIA (measured at 1 mg/mL total purified IgG), or an improved total antigen-specific GIA $EC_{50}$ versus RH5 alone; in fact, most antigen mixtures lacking RH5 and containing CyRPA and/or RIPR performed significantly worse (Fig. 4D, E). We thus further analysed rats immunised with the single full-length antigens across various doses. This confirmed a clear hierarchy of immuno-potency, with RH5 only requiring 64 µg/mL (95% CI: 50 – 89 µg/mL) antigen-specific rat IgG to achieve 50 % GIA, versus 183 µg/mL (95% CI: 165 – 204 µg/mL) for CyRPA and 715 µg/mL (95% CI: 669 – 778 µg/mL) for RIPR (Supplementary Fig. 4D–N). These data suggested that any potential for additivity or synergy, as highlighted by the mAb analyses, is almost certainly ablated by the relatively poor overall immuno-potency exhibited by the anti-RIPR polyclonal IgG response, which is further

compounded by its relative immuno-dominance in the antigen combination vaccines. Consequently, no new antigen combination could produce more GIA per µg of total antigen-specific antibody than that achieved by RH5 immunisation alone.

### Growth-inhibitory antibody epitopes in RIPR lie within EGF-like domains 5-8

Considering these data, and in an attempt to improve immunologic outcomes, we further investigated the antibody response to RIPR. This antigen was originally reported as a large 123 kDa protein containing 10 epidermal growth factor-like domains (EGF)[19], and a central polypeptide cleavage site[45] which divides the molecule into N- and C-terminal halves. More recent data have shown that structurally RIPR is divided into an N-terminal "RIPR core" region that spans up to EGF(4), whilst EGF(5-10) are present within the C-terminal "RIPR tail" region[24] (Fig. 5A). We initially expressed a series of RIPR protein fragments spanning most of the full-length RIPR sequence in HEK293 cells (Supplementary Fig. 5A). These were genetically fused to a monomeric Fc (monoFc) solubility domain[46] to improve expression, which could subsequently be removed via use of tobacco etch virus (TEV) protease

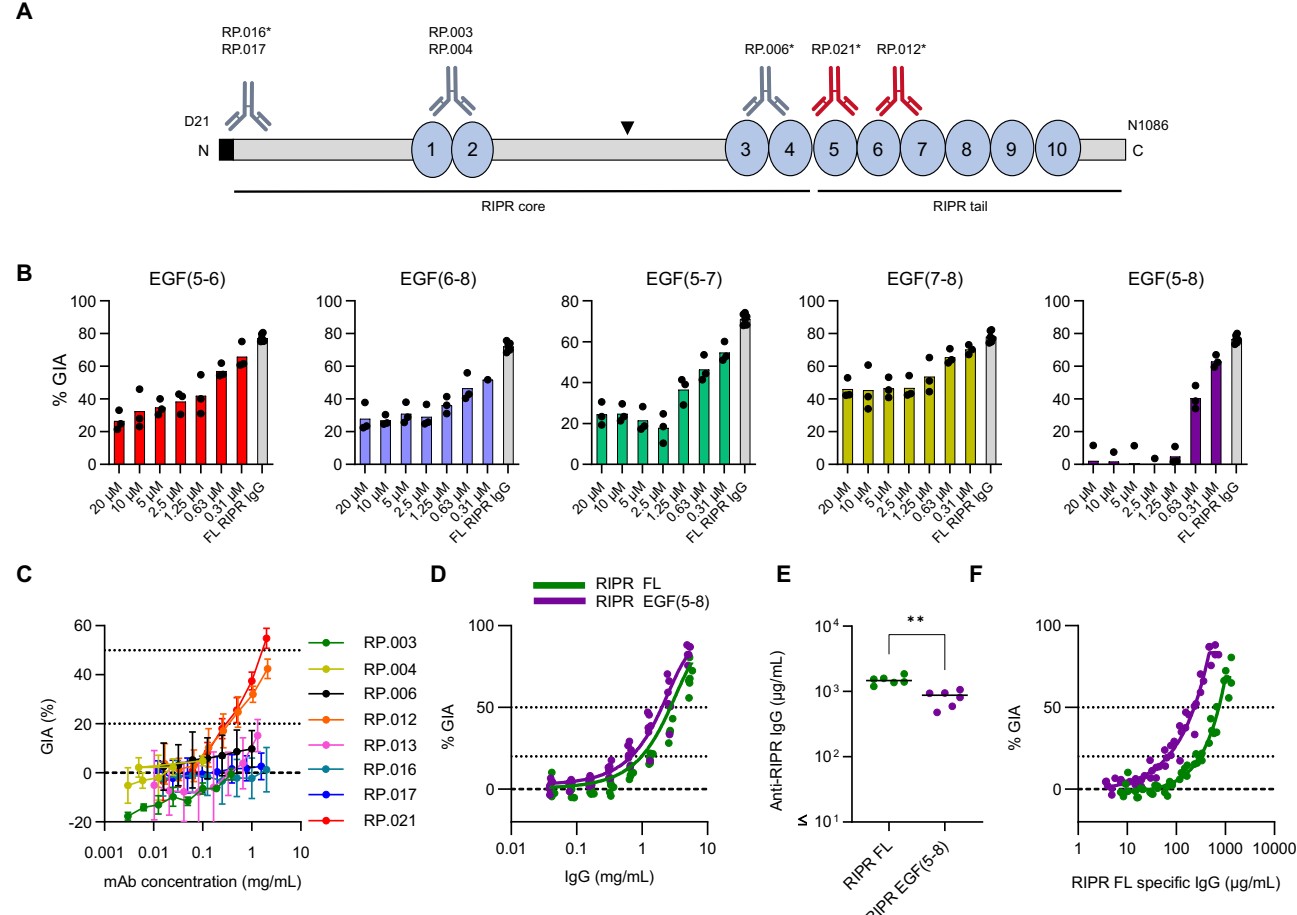

**Fig. 5 | EGF-like domains 5-8 of RIPR contain growth-inhibitory antibody epitopes. A** Schematic of the RIPR protein. Blue circles: EGF-like domains shown. Black triangle: PMX site[45]. Black bars: core and tail regions[24]. Anti-RIPR mAb binding sites; grey: GIA-negative mAbs; red: GIA-positive mAbs, asterisk indicates Type I mAbs. Type I mAb RP.013 could not be mapped. **B** Antigen reversal single-cycle GIA assay using *P. falciparum* clone 3D7 and pooled anti-RIPR IgG from rabbits (see Supplementary Fig. 5C). Anti-RIPR full-length (FL) total purified IgG was held at 3 mg/mL (grey bar) and the RIPR EGF proteins were titrated in the assay from 20 µM to 0.31 µM. Mean and data points (*n* = 3) are shown. **C** Eight anti-RIPR mAbs were titrated in a single-cycle GIA assay. GIA below the dotted line at 20 % is regarded as negative, the dotted line at 50 % GIA is included for clarity. Points are the mean of *N* = 3 replicates, error bars are the standard deviation (SD). **D** Wistar rats (*N* = 6/

group) were immunised with RIPR FL (green) or RIPR EGF(5-8) (purple) protein formulated in Matrix-M™ adjuvant. GIA assay was performed using total IgG purified from day 70 serum samples. Data from each rat were pooled and fitted with a Richard's five-parameter dose-response curve with no constraints. The dashed line shows 0 % GIA. The dotted lines show 20 % GIA and 50 % GIA for comparison. **E** Day 70 serum IgG ELISA data (reported in µg/mL) against full-length RIPR for rats immunised in (**D**). Dots represent individual animals and bars are median; ** *p* = 0.0022 by the two-tailed Mann-Whitney test. **F** Data from (**D**) replotted against RIPR FL-specific IgG concentration in µg/mL. Dashed line: 0 % GIA. Dotted lines: 20 % and 50 % GIA. Each data set was fitted with a Richard's five-parameter dose-response curve with no constraints and the $EC_{50}$ was calculated. Source data are provided as a Source Data file.

to yield a final panel of recombinant RIPR fragments (Supplementary Fig. 5B). Unlike the full-length RIPR protein, none of the RIPR fragments could be used to reconstitute the ternary RCR-complex with RH5 and CyRPA. To identify the regions of RIPR containing growth inhibitory epitopes, six rabbits were immunised with full-length RIPR to generate a large pool of GIA-positive anti-RIPR IgG (Supplementary Fig. 5C). We initially attempted to reverse the GIA of the anti-RIPR IgG by the addition of 1 µM of each RIPR protein fragment into the GIA assay. The only significant reversal was seen with the full-length RIPR positive control and RIPR EGF(7-8), with some non-significant reversal seen with two other proteins spanning RIPR EGF(5-6) or RIPR EGF(6-7) (Supplementary Fig. 5D). We therefore focused our efforts on the RIPR EGF(5-8) region and, using titrations of multiple RIPR EGF fragments, identified that only the RIPR EGF(5-8) protein was capable of complete GIA reversal (Fig. 5B and Supplementary Fig. 5E). These data strongly suggest this region is the sole target of growth-inhibitory antibodies in the induced polyclonal anti-RIPR IgG.

In light of these data, we hypothesised that the EGF(5-8) region of RIPR was not interacting with other binding partners in line with

GIA-positive epitopes identified on RH5 and CyRPA. Initially, we investigated the interaction of full-length RIPR (RIPR-FL) with semaphorin-7A (SEMA7A), a proposed binding partner of RIPR[47], however, we could not detect any binding between these proteins. We could, however, observe binding between another merozoite protein, MTRAP, and SEMA7A as previously reported[48] (Supplementary Fig. 6A) so we did not explore this interaction further. Subsequently, the *P. falciparum* PTRAMP-CSS heterodimer has been reported as a binding partner of RIPR; this interaction is mediated by the RIPR tail region, which includes EGF(5-8) and leads to the formation of the pentameric PCRCR-complex[24,25]. We therefore produced the PTRAMP-CSS heterodimer using a baculovirus expression system as previously described[25] (Supplementary Fig. C, D) and confirmed RIPR-FL binding to the PTRAMP-CSS heterodimer by SPR with a $K_D$ value of 4.5 µM (Supplementary Fig. 6E), highly similar to two previous reports[24,25]. However, we were unable to detect any binding between recombinant RIPR EGF(5-8) and the PTRAMP-CSS heterodimer (Supplementary Fig. 6F), suggesting this region of four EGF-like domains within the RIPR tail region is either insufficient or

not required to mediate this interaction and therefore likely to be exposed to neutralising antibodies.

We next sought to extend our GIA reversal data obtained using the polyclonal anti-RIPR rabbit IgG by epitope mapping the eight novel anti-RIPR mouse mAbs (Table S1). Here, we could identify binding regions within RIPR for 7/8 mAbs via dot-blot using the panel of RIPR protein fragments and a peptide array ELISA (Fig. 5A and Supplementary Fig. 5F, G). Consistent with our previous data, only two of the eight mAbs (RP.012 and RP.021; both Type I) showed GIA (Fig. 5C) and both had epitopes within RIPR EGF(5-8). In contrast, all the remaining mAbs bound elsewhere within the RIPR molecule and accordingly were GIA-negative. Considering these data, we next immunised rats with either 20 µg RIPR-FL or an equimolar amount of RIPR EGF(5-8) protein (3.97 µg). Both groups demonstrated equivalent overall GIA activity (Fig. 5D), however RIPR EGF(5-8) elicited significantly lower anti-RIPR serum IgG serum responses (Fig. 5E). Consequently, analysis of functional antibody quality (i.e., GIA per µg of anti-RIPR IgG) revealed a three-fold improvement, with RIPR EGF(5-8) lowering the $EC_{50}$ value to 232 µg/mL (95% CI: 219 – 246 µg/mL), down from 715 µg/mL (95% CI: 669 – 778 µg/mL) for the anti-RIPR-FL IgG (Fig. 5F).

### Design of RIPR(EGF)-CyRPA fusion protein vaccines

We hypothesised that replacing full-length RIPR with RIPR EGF(5-8) in an immunogen targeting the wider RCR-complex could reduce the immuno-dominance of RIPR whilst maintaining all the known Type I growth-inhibitory epitopes, at least one of which can also synergise with anti-RH5 antibodies. A pilot immunogenicity study using various proteins that span EGF(5-8) showed that low doses of these RIPR EGF domain proteins (< 0.5 µg soluble antigen, equivalent to 2 µg full-length RIPR) were not immunogenic for antibody induction, likely due to their relatively small size and/or lack of T cell help at low dose. We initially rescued these IgG responses by arraying the RIPR EGF protein fragments on hepatitis B surface antigen (HBsAg) virus-like particles[49] (VLPs) (Supplementary Fig. 7A). However, as an alternative strategy, we sought to simplify future immunogen manufacturing, whilst maintaining immunogenicity, by genetic fusion to CyRPA (Supplementary Fig. 7B, C). We were able to successfully express RIPR EGF(7-8) and RIPR EGF(5-8), each fused to CyRPA, and termed these new fusion proteins "R78C" and "R58C", respectively (Supplementary Fig. 7D); both proteins reacted with a panel of growth inhibitory anti-CyRPA mAbs suggesting correct conformation, in addition, RP.012 could only bind R58C in the same assay (Supplementary Fig. 7E). Furthermore, R58C could completely reverse the GIA from anti-RIPR full-length and anti-CyRPA rabbit IgG (Supplementary Fig. 7F, G) showing that all growth inhibitory epitopes from both RIPR and CyRPA are present in the R58C immunogen.

### Immunisation with R78C and RH5 gives improved GIA over RH5 alone

We next immunised groups of six rats with single soluble antigens (RH5, CyRPA, RIPR, RIPR EGF(7-8), R78C or R58C), or a mixture of R78C and RH5, or R58C and RH5. In addition, a 20 µg RH5 group was included to determine the effects of a higher RH5 dose. In the case of the mixtures, we assessed admixing the two proteins at the time of immunisation, as well as pre-formed binary complexes (Supplementary Fig. 7H, I), which we termed "RCR-78 mini" and "RCR-58 mini". All constructs were administered three times with 25 µg Matrix-M™ adjuvant. R78C and R58C immunogens were tested in separate studies due to antigen availability. The total anti-RH5, -CyRPA, and -RIPR serum IgG responses were measured by ELISA over time (Fig. 6A–F and Supplementary Fig. 8A). Encouragingly, following three doses, the anti-RH5 IgG response for all groups showed no significant reduction compared to RH5-only vaccinated animals, and there was no added benefit of using a higher 20 µg dose of RH5 over a 2 µg dose (Fig. 6A, B and Table S3). Following R78C + RH5 vaccination, anti-CyRPA IgG

responses were lower than those observed in the CyRPA-only vaccinated animals, but this did not reach significance. There was a significant reduction of the anti-CyRPA IgG response in the context of R58C and R58C + RH5 immunisation, as compared to the previous CyRPA-only vaccinated controls (Fig. 6C, D and Table S3). As expected, RIPR protein gave the highest anti-RIPR IgG response, and minimal immunogenicity was seen when immunising with soluble RIPR EGF(7-8) alone. However, all the R78C and R58C groups alone, or in combination with RH5, showed comparable responses, albeit significantly lower (5 to 10-fold) than those seen with full-length RIPR – consistent with antibodies only being raised against a much smaller portion of this molecule (Fig. 6E, F and Table S3). From these data, we concluded that the new R78C and R58C fusion protein constructs substantially reduced the immuno-dominance of RIPR, so that anti-RH5 IgG responses are unaffected by co-immunisation, and that these provided an immunogenic framework to deliver the small EGF domain targets of RIPR. The detrimental effect of co-immunisation on anti-CyRPA IgG responses was reduced with R58C in combination with RH5, and eliminated with R78C in combination with RH5, confirming that CyRPA is the least immunogenic antigen and that responses are sub-dominant when combined with the other antigens.

Purified IgG from each rat was subsequently tested for GIA against *P. falciparum* 3D7 clone parasites (Fig. 6G and Supplementary Fig. 8B). Animals immunised with soluble CyRPA, RIPR, RIPR EGF(7-8) and R58C performed significantly worse than RH5 alone in terms of overall GIA achieved at 1 mg/mL total IgG. Vaccination with 2 or 20 µg R78C also achieved a level of GIA that was lower than RH5 alone on average, which reached significance for the 2 µg dose group. When the R78C and R58C antigens were combined with RH5, the GIA from the R58C + RH5 combinations was comparable to RH5 alone. In contrast, the chimeric construct R78C + RH5 significantly outperformed RH5 alone suggesting that the shorter RIPR construct in R78C led to higher performance. The improvement seen here was also not due to the higher dose of RH5 protein used in the RH5 + R78C combination, given there was no difference observed in the anti-RH5 IgG response or GIA at 1 mg/mL total IgG when using either 2 or 20 µg RH5. Due to the superior performance of R78C over R58C when combined with RH5, further analyses were performed for R78C only.

We next compared the antibody quantity and quality produced by the different vaccine candidates across multiple studies to investigate why R78C in combination with RH5 was superior to both the full RCR-complex and RH5 alone. We sought to explain these results by first looking at the antigen-specific IgG response versus GIA (Supplementary Fig. 8C). This showed that the functional quality of CyRPA-, RIPR-, and RCR-based vaccines is worse than RH5 alone (in terms of the GIA assay $EC_{50}$). However, the quality of antibodies produced by R78C + RH5 vaccination maintained a similar overall functional quality to RH5 (Fig. 6H). In addition, there were higher levels of vaccine-induced antibodies in the R78C + RH5 combination vaccine groups (Fig. 6I) as compared to RH5 alone, with a reduction in the proportion of anti-RIPR IgG as compared to RCR-complex immunised groups (Fig. 6J). These data suggest that the significant improvement in overall GIA is due to an increased quantity of total antigen-specific antibodies in the R78C + RH5 vaccine groups whilst maintaining high functional potency.

### Anti-RH5, -CyRPA, and -RIPR IgG show additive GIA irrespective of immunisation strategy

We finally sought to confirm whether the polyclonal antibodies raised against the three RCR-complex antigens were acting additively or synergistically in the GIA assay. We initially combined polyclonal total IgG from rats immunised with single immunogens, and tested combinations of (i) CyRPA and RIPR; (ii) RH5, CyRPA and RIPR; and (iii) RH5 and R78C. In all cases, the level of GIA observed in the mixtures was highly comparable to the predicted level of GIA as defined by Bliss

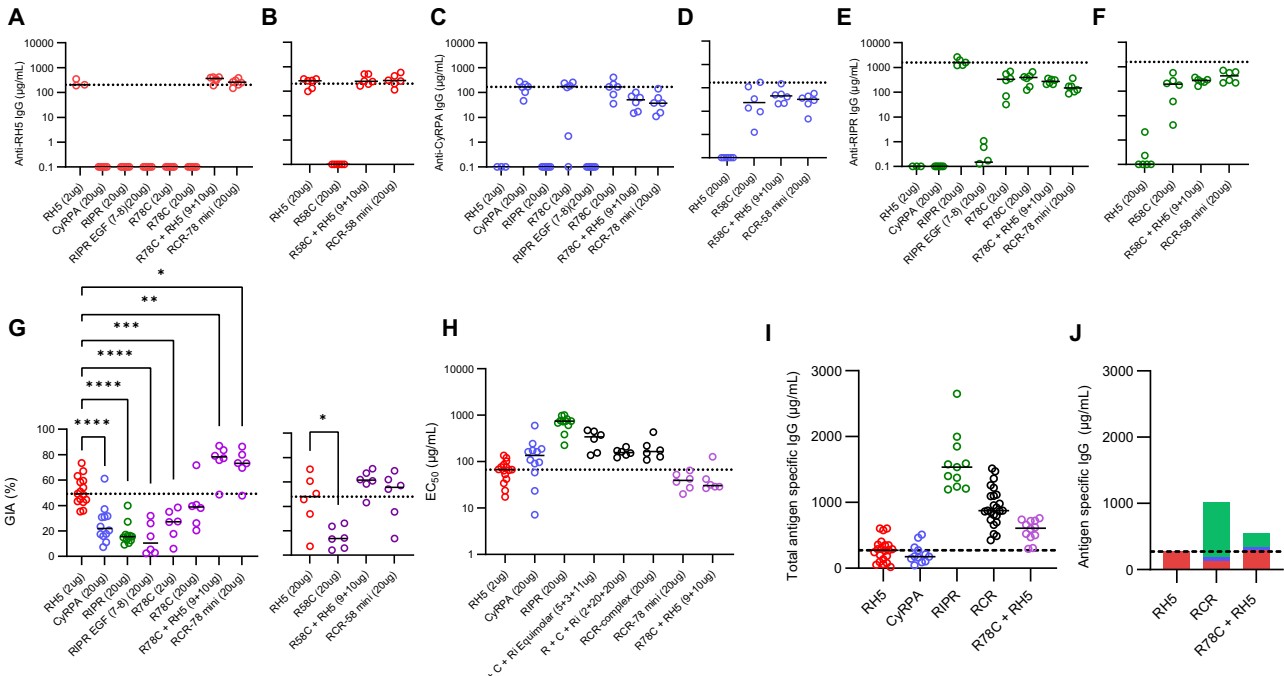

**Fig. 6 | Immunisation with combinations of RH5 and R78C improves on immunisation with RH5 alone.** Wistar rats were immunised on days 0, 28 and 56. A terminal bleed was taken on day 70 (post-third dose). Combination (e.g., "R78C + RH5") groups were given at equimolar ratios of 9 μg + 10 μg antigen respectively, whereas "mini" groups (e.g., "RCR-78 mini") were given as 20 μg of the pre-formed complex. Day 70 serum IgG ELISA data (reported in μg/mL) showed against full-length (**A, B**) RH5 (red), (**C, D**) CyRPA (blue), and (**E, F**) RIPR (green). Dotted line: median antigen-specific IgG response from a reference group of single antigen immunised animals. A summary of statistical analysis can be found in Table S3. *N* = 6 animals per group (**G**) Single-cycle GIA assays. Total IgG, purified from day 70 serum samples, was titrated in the GIA assay (see Supplementary Fig. 8B). GIA at 1 mg/mL total purified IgG was interpolated. The dotted line indicates the median GIA for the RH5 alone group. Individual and median group responses are shown. Data include all animals from previous studies vaccinated identically with RH5, CyRPA or RIPR at the indicated dose. Significance determined by one-way ANOVA

with Dunnett's multiple comparisons test versus RH5 only group, * *p* < 0.0104, ** *p* < 0.0017, *** *p* < 0.0005, **** *p* < 0.0001. RH5: *n* = 15, CyRPA: *n* = 12, RIPR: *n* = 11, all other groups *n* = 6 biologically independent experiments. **H** GIA EC$_{50}$ data from the indicated groups (from data in Supplementary Fig. 8C). The Dotted line indicates the median result for RH5-only immunised animals (2 μg dose) for comparison. RH5: *n* = 21, CyRPA: *n* = 12, RIPR: *n* = 11, RCR: *n* = 24, and R78C + RH5 *n* = 12 biologically independent experiments. **I** Combined day 70 serum IgG ELISA data (summed in μg/mL) for all three antigens across select vaccination groups receiving the same immunogens; 2 μg dose RH5 data is shown. The dashed line indicates the median result for RH5-only immunised animals for comparison. RH5: *n* = 15, CyRPA: *n* = 12, RIPR: *n* = 11, all other groups *n* = 6 biologically independent experiments. **J** Summary of data shown in (**I**) with the median contribution of each different antigen-specific IgG (in μg/mL) displayed for each immunisation group. RH5 (2 μg): red; CyRPA: blue; RIPR: green. The dashed line shows the median result for the RH5-only group for comparison. Source data are provided as a Source Data file.

additivity (Fig. 7A–C). We also assessed for potential interactions in the context of antigen co-immunisation by affinity purification of antigen-specific IgG from sera using single antigen (i.e., RH5, CyRPA or RIPR) affinity columns. Affinity-purified IgGs were then tested alone and combined. In all cases, and regardless of whether the IgGs were raised by immunisation with single antigens, the RCR-complex, or R78C + RH5, the test combinations showed levels of GIA that were equivalent to the predicted additive (Fig. 7D–F). Consequently, even though individual mAbs against these target antigens could display synergistic GIA, the polyclonal IgGs raised by these immunogens and specific formulations in rats, consistently demonstrated additive GIA.

## Discussion

RH5 was first reported in 2011 to be a promising antigen target for inclusion in a future blood-stage *P. falciparum* malaria vaccine[9,10]. Since then, a number of vaccine candidates based on the RH5 antigen have entered clinical development[12–14], with the most advanced (RH5.1/Matrix-M™) currently in a Phase 2b field efficacy trial. In parallel, a wealth of data has emerged surrounding the immuno-biology of RH5 and its presentation on the merozoite surface as part of a wider invasion protein complex[20,24]. Here, we sought to investigate whether a vaccine candidate based on the ternary RCR complex could substantially improve upon the leading clinical candidate RH5.1/Matrix-M™.

To guide this work, we initially explored the interaction of a panel of anti-RH5, -CyRPA and -RIPR mAbs with the recombinant RCR-complex. These analyses allowed us to divide the mAb panel into Type I (RCR-complex binding) or Type II (RCR-complex non-binding) clones and identified that all GIA-positive mAbs are Type I regardless of the target antigen, i.e., they have epitopes that are exposed on the surface of the formed RCR-complex. In contrast, GIA-negative mAbs could be either Type I or Type II. These results are consistent with available structural data for a subset of the anti-RH5 and -CyRPA mAb panels[6,24,36,37]. They also strongly suggest that preventing RCR-complex formation is not a mechanism of GIA and that this complex is likely formed within the parasite before any potential surface exposure to antibodies.

Having previously observed that certain anti-RH5 and anti-CyRPA mAbs can show intra-antigen synergistic GIA[36,37], we also explored the mAb panel for potential inter-antigen interactions using pair-wise combinations and representative clones from non-overlapping Type I epitope sites on each of the three antigens. As assessed by this GIA assay format, many combinations performed additively, however, we identified a clear propensity for two out of the four anti-RH5 mAbs to synergise with the anti-CyRPA and -RIPR clones. One of these mAbs, R5.011, has been reported previously and represents an anti-RH5 human antibody specificity that shows minimal or no GIA when tested alone but which can synergise in

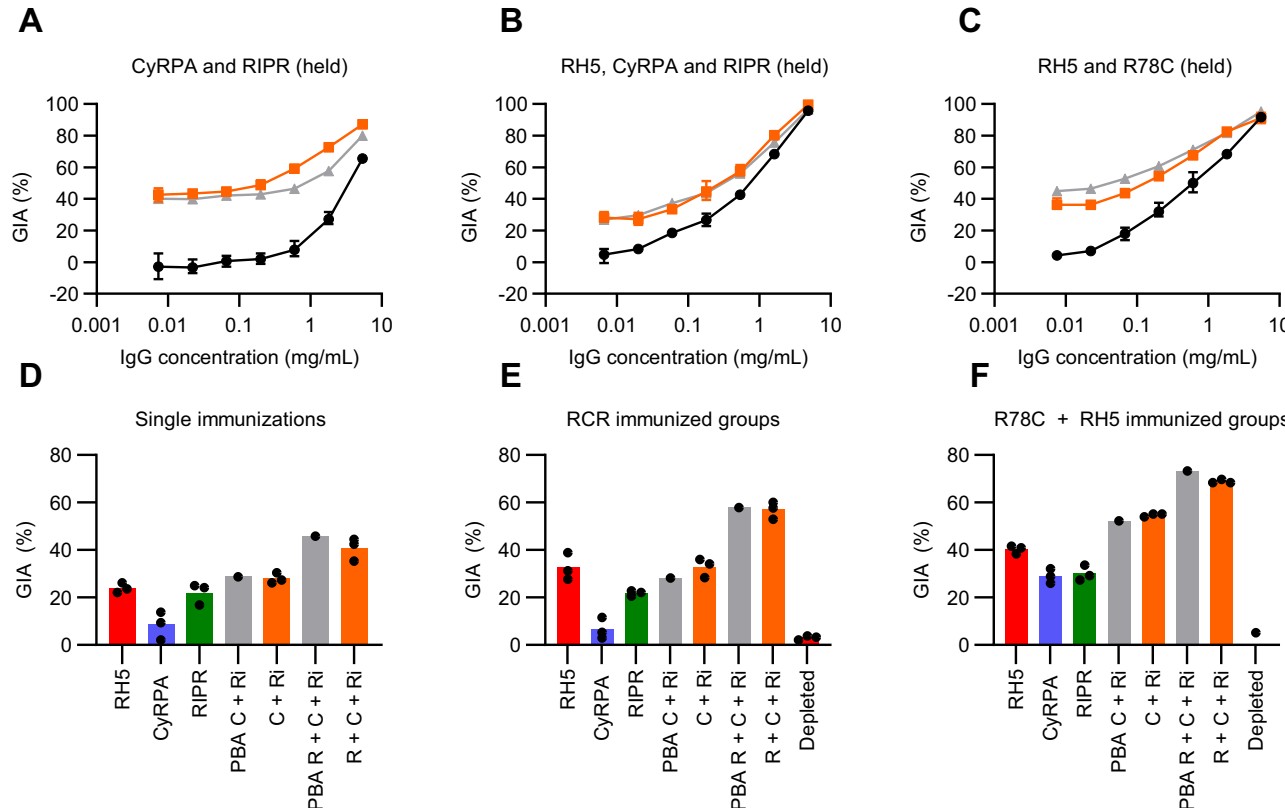

**Fig. 7 | Anti-RH5, -CyRPA, and -RIPR polyclonal vaccine-induced IgG show additive GIA.** Selected total IgG samples from rats vaccinated in Figs. 4 and 6 were titrated in a GIA assay using *P. falciparum* clone 3D7. **A** Anti-RIPR total IgG was held at 2.5 mg/mL (not shown, ~40 % GIA) with anti-CyRPA IgG titrated using a 5-fold dilution. Black: anti-CyRPA IgG alone; grey: Predicted Bliss Additivity (PBA) of the titrated anti-CyRPA IgG with the held anti-RIPR IgG; orange: measured result. Mean of *N* = 3 replicates, and SD shown. **B** Anti-RIPR and anti-CyRPA total IgG each held at 0.5 mg/mL (not shown, ~30 % GIA) with anti-RH5 IgG titrated using a 5-fold dilution series. Black: anti-RH5 alone; grey: PBA of the titrated anti-RH5 IgG with anti-CyRPA and anti-RIPR IgG held; orange: measured result. Mean of *N* = 3 replicates, and SD shown. **C** Anti-R78C total IgG was held at 1 mg/mL (not shown, ~30 % GIA) with anti-RH5 IgG titrated using a 5-fold dilution series. Black: anti-RH5 alone; grey: PBA of

titrated anti-RH5 IgG with the held anti-R78C IgG; orange: measured result. Mean of *N* = 3 replicates, and SD shown. **D–F** Single point GIA assay of purified antigen-specific IgG. Anti-RH5 (red), anti-CyRPA (blue), and anti-RIPR (green) each tested at a concentration aimed to give ~20 % GIA. "C + Ri": anti-CyRPA IgG + anti-RIPR IgG mix; "R + C + Ri": anti-RH5 + anti-CyRPA + anti-RIPR IgG mix. Antigen-specific IgG concentrations in each mix as per single antigen GIA experiments. PBA in grey; measured result in orange; depleted: post-purification IgG at 0.5 mg/mL. Mean of *N* = 3 technical replicates shown. Antigen-specific IgG purified from pooled sera from (**D**) single antigen immunised animals; (**E**) animals immunised with the RCR-complex; and (**F**) animals immunised with R78C + RH5. Source data are provided as a Source Data file.

combination with other growth inhibitory antibodies via reducing the speed of merozoite invasion[36]. The second clone, R5.008, is itself GIA-positive, with an epitope that overlaps the basigin binding site[36,50,51]. Interestingly, recent live cell imaging has confirmed the synergy of the R5.008+Cy.009 mAb combination and suggested that the inactivation of uninvaded parasites by these antibody combinations may function as a second inhibitory mechanism alongside the blockade of basigin receptor binding[52]. We did not observe inter-antigen synergy with R5.016, unlike previous reports[38], highlighting the complexity of synergy between antibodies against specific epitope regions of RH5 with anti-CyRPA and -RIPR clones. This remains an active area of further investigation as high-resolution epitope maps of CyRPA and RIPR are generated.

Nevertheless, these data indicated that growth inhibitory antibody epitopes are exposed on the formed RCR-complex and that antibody responses across the three antigens could act additively with some specificities also showing synergy. We therefore initially focussed on vaccination strategies using the formed RCR-complex, hypothesising that the polyclonal antibody response should be improved, as compared to the use of single antigens, via the masking of Type II epitopes. However, none of the vaccine candidates tested targeting the RCR-complex could outperform full-length RH5 (RH5.1) alone in terms of functional antibody induction in rats. This is consistent with

another recently reported study that attempted a similar strategy[53]. Our quantitative analysis of the antibody response to each antigen in μg/mL revealed this was due to both the immuno-dominance of RIPR (over RH5 and especially CyRPA) coupled with the relatively poor immuno-potency of anti-RIPR polyclonal IgG (in comparison to both anti-RH5 and -CyRPA polyclonal IgG). This latter observation was also consistent with the relatively poor immuno-potency of the anti-RIPR mouse mAb panel and the high proportion of GIA-negative clones within those classified as Type I.

We thus investigated the polyclonal antibody response to full-length RIPR in more depth. Here we identified by systematic GIA reversal assays that all of the growth inhibitory antibodies raised by full-length RIPR vaccination are located within an ~200 amino acid region of the RIPR tail corresponding to EGF(5-8). Consistent with this result have been previous reports by others highlighting EGF(5-8) as a target of growth-inhibitory antibodies in *P. falciparum*[19,47,54], along with our epitope mapping of the two GIA-positive mAbs reported here (RP.012 and RP.021). Indeed, recent cryo-EM structural data of RIPR[24] show that these EGF domains form part of the RIPR tail that extends out of the RIPR core towards the parasite membrane, consistent with our definition of these as Type I epitopes that are accessible to inhibitory mAbs within the context of the RCR-complex. We further showed that immunisation with RIPR EGF(5-8) could induce

comparable overall levels of GIA as full-length RIPR with a three-fold improvement in the antigen-specific $EC_{50}$.

We accordingly designed two new constructs based on this information: R78C and R58C; whereby we elected to fuse the small RIPR EGF domain region to CyRPA to maintain immunogenicity (as opposed to using a VLP scaffold) to both simplify antigen production and maintain focus on the RCR-complex antigenic components. Co-immunisation of these new constructs with RH5.1 reduced the immuno-dominance of the RIPR component and focused the response on the most potent RIPR epitopes, as anticipated, thereby maintaining the anti-RH5 IgG response, and reducing interference with the sub-dominant anti-CyRPA IgG response. Notably, immunisation with the combination of R78C and RH5.1 led to a significant improvement in overall GIA (as compared to RH5.1 alone). Our analysis indicated this was due to an increase in the overall total quantity of antigen-specific IgG (in the R78C + RH5.1 vaccinated rats) with similar functional potency (or quality) as compared to RH5.1 vaccination alone.

Finally, we demonstrated that the antigen-specific rat IgGs, induced by vaccination with the three RCR-complex antigens, inter-act in an additive manner in the GIA assay. Consequently, although we could define specific synergistic inter-antigen interactions with the mAb panels used here, we could not replicate this with polyclonal IgG responses induced by vaccination. Potential reasons for this could include species specificity of the induced antibody repertoire to each antigenic component; much greater complexity of these interactions within polyclonal mixtures, and/or insufficient induction of synergising antibody specificities within the polyclonal responses. Linked with this we also found, somewhat surprisingly, the R58C vaccine candidate performed less well in combination with RH5.1 (as compared to R78C). This may indicate the most effective anti-RIPR epitopes for combination with RH5 and CyRPA lie within EGF(7-8), however, the mechanism(s) by which such antibodies function remains to be determined. Indeed, we could not detect any binding between recombinant RIPR and SEMA7A as previously reported[47]. Instead, recent structural data have shown the RIPR tail region (spanning from EGF(5) to the end of the C-terminal domain [CTD]) interacts with the PTRAMP:CSS heterodimer as part of the wider PCRCR-complex[24]. However, our data indicate the smaller EGF(5-8) region is either insufficient or not required to mediate this interaction, suggesting that blockade of RIPR binding by these antibodies to PTRAMP:CSS on the merozoite surface is unlikely; this is also in line with our data that show that blockade of RCR-complex formation is not an inhibitory mechanism. Further studies with much larger panels of GIA-positive anti-RIPR mAbs are now required to fine-map the most potent RIPR epitopes and determine the mechanism(s) of antibody-mediated inhibition to provide a higher-resolution framework that could guide a more focused vaccine design.

In conclusion, the combination of R78C + RH5.1 in Matrix-M™ adjuvant is the first vaccine candidate based on the wider RH5 invasion complex to show a significant improvement in overall GIA as compared to RH5.1/Matrix-M™ in preclinical studies. The R78C antigen has since completed biomanufacture in line with current Good Manufacturing Practice (cGMP) and has now entered a Phase 1a clinical trial in the United Kingdom (ClinicalTrials.gov NCT05385471) formulated either alone or in combination with RH5.1 in Matrix-M™ adjuvant. For the Phase 1a trial, R78C, RH5.1, and Matrix-M™ will be admixed at the bedside immediately before immunisation rather than producing a "RCR-78 mini" vaccine. This was chosen due to the absence of any significant difference between the mixed and "RCR-78 mini" groups reported here, and to simplify manufacturing and regulatory requirements. This will be the first assessment in humans of the safety and immunogenicity of a novel vaccine candidate targeting the wider RCR complex and will enable future studies to link human anti-CyRPA and anti-RIPR immune responses to functional anti-parasitic and vaccine efficacy outcome measures.

## Methods

### Recombinant protein expression and purification

The recombinant full-length RH5, CyRPA, RIPR, CSS, and PTRAMP protein sequences were all based on the 3D7 clone *P. falciparum* reference sequence. RH5 encoded amino acids E26-Q526 as published previously (and called "RH5.1")[17] and four mutations to remove N-linked glycosylation sequons: T40A, T216A, T286A and T299A. CyRPA encoded amino acids D29-E362 with three mutations to remove N-linked glycosylation sequons: S147A, T324A, and T340A[37]. RIPR encoded amino acids M1-N1086, with 12 mutations to remove N-linked glycosylation sequons: N103Q, N114Q, N228Q, N334Q, N480Q, N498Q, N506Q, N526Q, N646Q, N647Q, N964Q, N1021Q. Each protein construct included an N-terminal secretion signal and a C-terminal four-amino acid purification tag[17,39] (C-tag: EPEA).

RH5 and RIPR proteins were expressed as secreted proteins by stable polyclonal *Drosophila* S2 cell lines (ExpreS²ion technologies) as previously reported[17,40]. CyRPA proteins were expressed as secreted protein by transient transfection of HEK Expi293 cells (Thermo Fisher Scientific) following the manufacturer's protocol using ExpiFecta-mine™ (Thermo Fisher Scientific). All supernatants were harvested via centrifugation and the proteins were purified using CaptureSelect C-tag affinity matrix (Thermo Fisher Scientific) on an ÄKTA Pure FPLC system (Cytiva). A further size exclusion chromatography (SEC) polishing step was performed on a HiLoad 16/60 Superdex 200 pg column (GE Healthcare) in 20 mM Tris, 150 mM NaCl, pH 7.4.

After purification of full-length RH5, CyRPA and RIPR, the RCR-complex was produced by mixing equimolar concentrations of each protein and incubating for 30 min at room temperature (RT). The assembled complex was then purified by SEC using an S200 16/600 column and ÄKTA Pure (Cytiva) into Tris-buffered saline (TBS).

Recombinant SEMA7A and MTRAP were produced in and purified from HEK Expi293 cells as described above for full-length CyRPA. The SEMA7A and MTRAP plasmids were a kind gift from Gavin Wright[48] (Addgene plasmid #73115 and #47746). Plasmid #73115 was digested with NotI and EcoRI to excise the SEMA7A sequence, before ligation into a modified pENTR4LP[55] vector containing an in-frame C-terminal C-tag. The MTRAP sequence was cloned form #47746 by PCR into a modified pENTR4LP vector containing an in-frame C-terminal C-tag without any further solubility domain.

CSS, encoding amino acids G20-K290 with N-linked glycosylation sequons intact, and PTRAMP, encoded amino acids C42-T309 with one mutation to remove an N-linked glycosylation sequon (T197A), as previously reported[25] were subcloned into the dual promoter pOET5.1 transfer vector (Oxford Expression Technologies). A biotin acceptor peptide (BAP) tag was included at the C-terminal end of PTRAMP before the C-tag. The flashBAC gold™ system (Oxford Expression Technologies) was used to recombinantly express the secreted PTRAMP-CSS heterodimer along with biotin ligase (BirA) in Sf9 cells. The proteins were purified via CaptureSelect™ C-tag affinity matrix (Thermo Fisher Scientific). A further SEC polishing step was performed on a HiLoad 16/600 Superdex 200 pg column (GE Healthcare) in Dulbecco's PBS (DPBS). Each protein was verified using SDS-PAGE, western blot using ExtrAvidin® alkaline phosphatase (Sigma-Aldrich) to detect the BAP tag, liquid chromatography-tandem mass spectrometry (LC-MS/MS), and intact mass analysis. Mass spectroscopy was performed at the Centre for Medicines Discovery (CMD), University of Oxford.

RIPR protein truncations (unless otherwise stated) were cloned from the full-length RIPR gene template by PCR (Table S4). Primers were designed with 3' BamHI and 5' KpnI sites flanking each sequence and the fragment was amplified using Phusion™ High-Fidelity DNA Polymerase (Thermo Fisher Scientific). The purified PCR product was digested with BamHI-HF and KpnI-HF (NEB) and ligated (Quick Ligase, NEB) into a modified pENTR4LP[55] vector containing the monomeric Fc ("monoFC") solubility domain[46] and a mouse IgG signal sequence. Proteins were expressed as secreted protein by transient transfection

of HEK Expi293 cells (Thermo Fisher Scientific) and purified via CaptureSelect™ C-tag affinity matrix (Thermo Fisher Scientific). A further SEC polishing step was performed on a HiLoad 16/600 Superdex 200 pg column (GE Healthcare) in 20 mM Tris, 150 mM NaCl, pH 7.4. Each protein was verified using SDS-PAGE, liquid chromatography-tandem mass spectrometry (LC-MS/MS), and intact mass analysis. Mass spectroscopy was performed at the Centre for Medicines Discovery (CMD), University of Oxford.

The RIPR N-half fragment (Table S5) was generated using the 3D7 clone *P. falciparum* sequence with N-linked glycosylation sequons left intact, the monoFc and C-tag. The RIPR N-half was produced in HEK293F cells using the Expi293™ Expression System (Thermo Fisher Scientific) in the presence of 5 µM kifunensine (Abcam). Following affinity purification with CaptureSelect™ C-tag affinity matrix (Thermo Fischer Scientific), the purified protein was dialysed overnight in TBS, with a 1:50 ratio of purified protein to Endoglycosidase H (Endo H) (Promega). After treatment, the Endo H was removed on a Superdex™ 200 Increase 10/300 GL size exclusion column (GE Healthcare).

To remove the monoFc on relevant proteins, 5 mg purified RIPR protein fragment was incubated with TEV protease (Sigma/Promega) at a 1:10 v/v ratio overnight at 4 °C on a rolling mixer. The sample was then centrifuged at 10,000 x$g$ in a benchtop centrifuge before affinity purification using a 1 mL CaptureSelect™ C-tag affinity matrix followed by a Superdex™ 75 Increase 10/300 GL SEC column (Cytiva).

To generate virus-like particle vaccines: hepatitis B surface antigen (HBsAg) fused to the SpyCatcher[49] peptide was incubated on ice with an equimolar ratio of RIPR EGF (5-8), (5-6), or (7-8) fused to the SpyTag for 20 min to allow conjugation to occur. The mixture was then purified on a Superose 6 increase 10/300 GL column (Cytiva) to remove unconjugated RIPR protein.

The R78C and R58C fusion protein sequences were based on the full-length CyRPA and RIPR sequences used above (Table S6). The R78C construct consists of amino acids D817-V897 of RIPR, two GGSGS linkers, and the SpyTag peptide[56]; to enable the production of VLPs if required two GGGGS linkers, amino acids D29-E362 of CyRPA and the C-tag at the C-terminus. R58C is the same construct except with amino acids P716-D900 of RIPR. The reverse order of antigens was not explored since both R78C and R58C expressed at useable levels in HEK Expi293 cells. R78C and R58C were expressed as secreted proteins by transient transfection of HEK Expi293 cells (Thermo Fisher Scientific). Proteins were purified via CaptureSelect™ C-tag affinity matrix (Thermo Fisher Scientific). A further SEC polishing step was performed on a HiLoad 16/60 Superdex 200 pg column (GE Healthcare) in 20 mM Tris, 150 mM NaCl, pH 7.4. "R78C-mini" and "R58C-mini" vaccines were produced by mixing equimolar concentrations R78C and RH5, or R58C and RH5, and incubating for 30 min at room temperature (RT). The assembled complex was then purified by SEC using an S200 16/600 column and ÄKTA Pure (Cytiva) into Tris-buffered saline (TBS).

## Antibody expression and purification

Generation of anti-RH5 and anti-CyRPA recombinant and chimeric mAbs (Table S1) has been previously described[16,36,37,43,57]. These mAbs were transiently expressed in Expi293F HEK cells. Cognate heavy and light chain-coding plasmids were co-transfected at a 1:1 ratio. Supernatants were harvested via centrifugation. All mAbs were purified using a 5 mL Protein G HP column (Cytiva) on an ÄKTA Pure FPLC system (Cytiva). Equilibration and wash steps were performed with PBS and mAbs were eluted in 0.1 M glycine pH 2.7. The eluates were pH equilibrated to 7.4 using 1.0 M Tris HCl pH 9.0 and immediately buffer exchanged into DPBS and concentrated using an Amicon ultra centrifugal concentrator (Millipore) with a molecular weight cut-off of 30 kDa.

Total IgG from rat serum was purified on drip columns packed with Pierce Protein G agarose resin (Thermo Fisher Scientific). Pierce Protein G IgG binding buffer (Thermo Fisher Scientific) was used to dilute the serum 1:1 before loading as well as for equilibration and wash steps. Bound IgG was subsequently eluted, neutralised and concentrated as for mAbs above.

## Analysis of mAb binding to the RCR-complex

The three components of the RCR-complex were pre-incubated at equimolar amounts, equivalent to 30 µg RH5, for 20 min to allow the RCR-complex to form. An equimolar amount of test mAb was then added and the sample incubated for 20 min. Each sample was then run on a Superdex 200 Increase size exclusion column (Cytiva). A 10 µL sample of each peak was collected and analysed on a non-reducing SDS-PAGE gel to ascertain which proteins were present in each peak (NuPAGE™ 4 to 12%, Bis-Tris, 1.0-1.5 mm, midi Protein Gels (Invitrogen) run at 200 V for 45 min).

## RCR-complex and test mAb immunoprecipitation

RH5, CyRPA and RIPR were mixed at an equimolar ratio and incubated for 30 min at RT to form the RCR complex. 40 µL 1 µM RCR-complex was then mixed with 100 µL mAb at 0.5 µM. A no-test mAb control of 40 µL 1 µM RCR-complex with 40 µL TBS was included in each run. After 10 min incubation, 30 µL protein G resin was added to each tube followed by a final 10 min incubation. Each tube was then washed 5 times with TBS pH 7.4 containing 10 % glycerol and 0.2 % Igepal C630. Following washing, the resin was resuspended in 1X sample buffer and briefly incubated at 100 °C. A 10 µL sample was then analysed by non-reducing SDS-PAGE as described above.

## Assay of growth inhibition activity (GIA)

GIA assays against 3D7 clone *P. falciparum* parasites was carried out over one blood-stage growth cycle ( ~ 48 h) as previously described[37,58]. Briefly: The assay was performed at indicated concentrations of purified total IgG or mAb in duplicate wells and a biochemical measurement using *P. falciparum* lactate dehydrogenase assay was used to quantify parasitaemia and define % growth inhibition. To ensure consistency between experiments, in each case, the activity of a negative control mAb, EBL040[59], which binds to the Ebola virus glycoprotein, and three anti-RH5 mAbs with well-characterised levels of GIA (2AC7, QA5, 9AD4)[57] were run alongside the test samples and used for assay QC. Purified total IgG samples from immunised animals were pre-incubated with human group O RhD-positive RBC to eliminate spurious GIA results caused by hemagglutination. Test antibodies were buffer exchanged into incomplete parasite growth media (RPMI, 2 mM L-glutamine, 0.05 g/L hypoxanthine, 5.94 g/L HEPES) before performing the GIA assay.

To assess for synergistic GIA, pairs of mAbs or purified total IgGs were assessed by measuring: i) the GIA of one test sample held at constant concentration to give approximately 20–40 % GIA; ii) the GIA of a second test sample (either total IgG from serum or mAb) titrated typically across a four-fold seven-step dilution curve; and iii) the GIA of the combination of the first sample held at a constant concentration combined with the second sample across its dilution curve. The predicted Bliss additivity was determined based on the measured activity from each antibody alone using formulas previously described[30]. Fold increase was determined by dividing the observed GIA by the predicted Bliss additivity.

For GIA-reversals, the experiment was carried out as above using a fixed amount of antigen-specific IgG. In addition, purified full-length proteins or protein fragments were titrated into the assay with the IgG at a defined concentration, typically between 0.5 and 20 µM. Each protein was tested in the absence of test IgG as a control.

All the blood donations and purchases at the University of Oxford for use in the GIA assay are anonymised and covered under ethical approval from the National Services of Health (REC reference 18/LO/0415, protocol number OVC002).

## Standardised ELISAs

ELISAs were performed against full-length RH5, CyRPA, or RIPR protein using standardised methodology as previously described[12,60]. AP-conjugated anti-rat IgG (A8438, Sigma-Aldrich) was used as a secondary antibody. A standard curve and Gen5 ELISA software v3.04 (BioTek, UK) were used to convert the optical density 405 nm ($OD_{405}$) of individual test samples into arbitrary units (AU). These responses in AU are reported in µg/mL following generation of a conversion factor by calibration-free concentration analysis (CFCA) as reported previously[12]. For the R + C + R ELISA, an equimolar ratio of RH5, CyRPA, and RIPR was used to coat the ELISA plate, the same secondary antibody and development conditions were used.

For endpoint ELISAs against R78C and R58C, Nunc™ MaxiSorp™ plates were coated with R78C or R58C at 2 µg/mL overnight. Plates were blocked with Blocker™ Casein (ThermoFisher). Primary antibodies were diluted to 2 µg/mL in PBS containing 1 mM $CaCl_2$. AP-conjugated anti-rat IgG (A8438, Sigma-Aldrich) was used as a secondary at 1:2000. Plates were developed with pNPP alkaline phosphatase with diethanolamine substrate and optical density read at 405 nm on an Infinite 50 plate reader (Tecan).

## Rabbit immunisations

Rabbits were immunised in two cohorts. The first was performed by Cambridge Research Biochemicals (CRB, Billingham, UK) in compliance with the UK Animals (Scientific Procedures) 1986 Act (ASPA). Four female 16-week-old New Zealand white rabbits were immunised by the intramuscular (IM) route with 20 µg RIPR antigen formulated in complete Freund's adjuvant on day 0 and incomplete Freund's adjuvant on days 14, 28, and 42. Pre-immunisation bleeds were taken at day −2 and final bleeds were taken at day 56. The second study was performed by Noble Life Sciences (Woodbine, MD, USA, which was AALACi accredited and OLAW assured) using female 16-week-old New Zealand white rabbits. Two rabbits were immunised IM with 50 µg RIPR antigen on days 0, 21 and 42 in Montanide ISA720 adjuvant (Seppic). Pre-immunisation bleeds were taken at day 0 and final bleeds were taken at day 64. Sera from both rabbit experiments were combined for the reported analyses.

## Rat immunisation studies

Rat immunisations were performed by Noble Life Sciences, Inc (Woodbine, MD, USA) using 8-12-week-old female Wistar rats (150–200 g). Groups of 6 rats were immunised IM with antigen (2 µg RH5, other antigens 20 µg unless otherwise stated) formulated in 25 µg Matrix-M™ adjuvant (Novavax) on days 0, 28 and 56. Tail bleeds were taken on days −2, 14 and 42. Final bleeds were taken on day 70.

## Purification of antigen-specific IgG

Recombinant RH5, CyRPA, or RIPR protein were coupled to HiTrap NHS-Activated HP affinity columns (Cytiva) using standard amine coupling protocols. Antigen-specific IgG was purified from polyclonal total IgG purified from serum using each antigen column on an ÄKTA Pure FPLC system (Cytiva). IgG was eluted in 0.1 M glycine, pH 2.7, followed by pH equilibration to 7.4 using 1.0 M Tris HCl, pH 9.0, and immediately buffer-exchanged into incomplete parasite growth media (RPMI, 2 mM L-glutamine, 0.05 g/L hypoxanthine, 5.94 g/L HEPES) and concentrated using an Amicon ultra centrifugal concentrator (Millipore) with a molecular weight cut-off of 30 kDa.

## RIPR monoclonal antibody production

Female 6-week-old BALB/c mice ($N = 4$ per group) were immunised IM with 10 µg N-half RIPR or 13 µg full-length RIPR protein formulated in AddaVax™ (Invivogen, France) on days 0, 14, 28 and 42. Spleens were harvested on day 56 and processed using EasySep™ Mouse B Cell Isolation Kit (StemCell). Hybridomas were generated by the fusion of B cells with SP2/0 cells (ATCC: CRL-1581) using ClonaCell™-

HY Hybridoma Kit (StemCell) using the manufacturer's instructions. Successful hybridomas were cultured in CELLine 1 L classic bioreactor flasks (Integra biosciences) in DMEM (Sigma-Aldrich) supplemented with 2 mM L-glutamine, 100 U/mL penicillin, 0.1 mg/mL streptomycin and 10 % ultra-low IgG foetal bovine serum (Thermo Fisher Scientific). mAbs were purified using a 5 mL Protein G HP column as described above. Mice were kept on a 12-12 light cycle with a phased dusk and dawn period. The temperature and humidity is kept within 21 °C ( +/− 2 °C) and 55% ( +/− 10 %). Procedures on mice were performed in accordance with the UK Animals (Scientific Procedures) Act Project Licence PPLs PA7D20B85 and PP7770851 and were approved by the University of Oxford's Animal Welfare and Ethical Review Body.

## RP.012 monoclonal antibody production

Five 6-week-old female SJL mice were immunised by Precision Antibody with full-length RIPR protein prepared by ExpreS²ion Biotechnologies, Denmark. Immunised mice were tested for titre by tail bleed ELISAs. Additional boosts were performed until sufficiently high tail bleed titres were observed. Based on titer data, a single mouse was selected for performing hybridoma fusions. Following fusions, monoclonal hybridoma screening was performed using hybridoma culture supernatants. Selected clone candidates were expanded for supernatant harvest and cryopreservation. Clone 3H7, now called RP.012, was selected and taken forward for further testing.

## Surface plasmon resonance (SPR)

SPR was carried out using the Biacore™ X100 machine and software. RIPR, SEMA7A, or PTRAMP-CSS were immobilised onto separate CM5 Sensor Chips (Cytiva) using the standard amine coupling protocol, yielding ~900 response units (RU) for each antigen. CyRPA was diluted in SPR buffer (PBS + P20: 137 mM NaCl, 2.7 mM KCl, 10 mM $Na_2HPO_4$, 1.8 mM $KH_2PO_4$, 0.005 % surfactant P20 (Cytiva)) to yield a top concentration of 30 µM; SEMA7A was diluted to a final concentration of 20 µM; MTRAP to a final concentration of 30 µM; RIPR to a final concentration of 6 or 2 µM; and RIPR EGF (5-8) to a final concentration of 4 µM. For each protein, a two-fold serial dilution series was then prepared in the same buffer. Samples were injected for 180 s at 30 µL/min before dissociation for 700 s. The chip was regenerated with a 30 s injection of 10 mM glycine pH 1.5, or 20 mM NaAc 100 mM NaCl pH 4.0 for PTRAMP-CSS CM5 chips. Data were analysed using the Biacore X100 Evaluation software v2.0.2, and the equilibrium dissociation constant ($K_D$) was determined from a plot of steady-state binding levels.

## Dot blots

200 ng test protein was blotted onto a nitrocellulose membrane and allowed to dry. Dot blots were then performed with the iBind™ Western Device (Thermo Fisher) with the test mAb diluted to 5 µg/mL and AP-conjugated goat anti-mouse IgG (A3562 Sigma-Aldrich) used as a secondary.

## RIPR peptide ELISA

A set of sixty-two biotinylated 20-mer peptides of RIPR overlapping by 12 amino acids, corresponding to amino acids D21-P247 and K364-S648 of RIPR, were synthesised (Mimotopes). Peptides were resuspended in 1 mL dimethyl sulfoxide and diluted to 5 µg/mL in DPBS for coating streptavidin-coated 96-well plates (Pierce). Plates were washed with PBS + 0.05% Tween 20 and then blocked with 200 µL Blocker™ Casein (Thermo Fisher). After washing, the test mAb was added at an initial concentration of 2 µg/mL. AP-conjugated goat anti-mouse IgG (Thermo Fisher) was used as a secondary and plates were developed with pNPP alkaline phosphatase with diethanolamine substrate and optical density read at 405 nm on an Infinite 50 plate reader (Tecan).

## Quantification and statistical analysis

Data were analysed using GraphPad Prism version 9.5.1 for Windows (GraphPad Software Inc.). Statistical tests are used and post-tests for multiple comparisons are reported in the Legends. In all statistical tests, reported $p$ values are two-tailed with $p < 0.05$ considered significant.

## Reporting summary

Further information on research design is available in the Nature Portfolio Reporting Summary linked to this article.

## Data availability

All data supporting the findings of this study are available within this paper and its Supplementary Files. Source data are provided with this paper. Further information and requests for resources should be directed to and will be fulfilled by the Lead Contact, Simon J. Draper (simon.draper@bioch.ox.ac.uk). Source data are provided with this paper.

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

## Acknowledgements

The authors are grateful for the assistance of Julie Furze, Penelope Lane, Wendy Crocker, Charlotte Hague, Rebecca Ashfield, Jee-Sun Cho, Geneviève Labbé, Carolyn Nielsen, Martino Bardelli, Francesca Donnellan, Gaurav Gupta, Yu Zhou, Yuanyuan Li, Sumi Biswas, Jenny Bryant and Lana Strmecki (University of Oxford); Sally Pelling-Deeves for arranging contracts (University of Oxford); Rod Chalk and Tiago Moreira (Centre for Medicines Discovery) for performing the mass spectroscopy analysis; Alex Cook, Sam Chamberlain and Brendan Farrell (University of Oxford) for their helpful discussions and SPR expertise; Ann-Marie Shorrocks (University of Oxford) for help with the immunoprecipitation experiments; Adéla Nacer and Paul Bowyer (NIBSC, UK) for provision of reagents; Jenny Reimer (Novavax); Ken Tucker, Timothy Phares and Cecille Browne (Leidos); and Robin Miller (USAID). This work was funded in part by the European Union's Horizon 2020 research and innovation programme under a grant agreement for OptiMalVax (733273); the UK MRC Confidence in Concept (CiC) Tropical Infectious Disease Consortium [MC_PC_17167]; and the Infectious Disease Division, Bureau for Global Health, United States Agency for International Development (USAID), under the terms of the Malaria Vaccine Development Program (MVDP) (AID-OAA-C-15-00071, for which Leidos, Inc. was the prime contractor, and 7200AA20C00017, for which PATH is the prime contractor). The GIA assays were supported in part by the Division of Intramural Research of the National Institute of Allergy and Infectious Diseases (NIAID), National Institutes of Health (NIH), and by an Interagency Agreement (AID-GH-T-15-00001) between the USAID MVDP and NIAID, NIH. The opinions expressed herein are those of the authors and do not necessarily reflect the views of USAID. This work was also supported in part by the National Institute for Health Research (NIHR) Oxford Biomedical Research Centre (BRC) and NHS Blood & Transplant (NHSBT; who provided material), the views expressed are those of the authors and not necessarily those of the NIHR or the Department of Health and Social Care or NHSBT. Cy.003, Cy.004, Cy.007, and Cy.009 were provided by Icosagen AS through the Centre for AIDS Reagents repository at the National Institute for Biological Standards and Control, UK. These mAbs were produced through the European Commission FP7 EURIPRED project (INFRA-2012-312661), funded by the European Union's Seventh Framework Programme [FP7/2007-2013] under Grant Agreement No: 312661-European Research Infrastructures for Poverty Related Diseases (EURIPRED). BGW held a UK MRC PhD Studentship [MR/N013468/1]. RJR and JJI were funded by the Wellcome Trust Infection, Immunology and Translational Medicine DPhil programme [105399/Z/14/Z]; DGWA held a UK MRC iCASE PhD Studentship [MR/K017632/1]; ADD held a Wellcome Trust Early Postdoctoral Research Training Fellowship for Clinicians [201477/Z/16/Z]; MKH held a Wellcome Trust Investigator award [220797/Z/20/Z]; ADD and SJD are Jenner Investigators and SJD held a Wellcome Trust Senior Fellowship [106917/Z/15/Z].

## Author contributions

Conceived and performed experiments and/or analysed the data: B.G.W., L.D.W.K., D.P., D.Q., A.M.L., S.E.S., R.J.R., H.D., J.R.B., K.M.H.,

C.A.R., L.B., M.W.S., L.A.C., R.A.D., A.D.D., O.R.L., K.M.H., M.K.H., and S.J.D. Contributed reagents, materials, and analysis tools: R.J.R., D.G.W.A., R.A.D., D.J.P., J.J.I., J.J., C.C., V.K., K.M., and C.A.L. Performed project management: J.M.C., A.R.N., R.S.M., C.R.K., A.J.B., L.A.S., and K.S. Wrote the paper: B.G.W., and S.J.D.

## Funding

This research was funded in part by the UK Medical Research Council (MRC) [Grant numbers: MC_PC_17167 and MR/N013468/1]. For the purpose of Open Access, the author has applied a CC BY public copyright licence to any Author Accepted Manuscript (AAM) version arising from this submission.

## Competing interests

S.J.D. is an inventor on patent applications relating to RH5 or RCR-complex malaria vaccines and antibodies and is a co-founder of and shareholder in SpyBiotech. B.G.W., L.D.W.K., D.P., D.Q., A.M.L., S.E.S., J.R.B., K.M.H., D.G.W.A., A.D.D., J.J.I., and M.K.H. are inventors on patent applications relating to RH5 and/or RCR-complex malaria vaccines and/or antibodies. J.J. is an inventor of patent applications relating to vaccines made using spontaneous amide bond formation and is a co-founder of and shareholder in SpyBiotech. R.A.D. is an inventor of patent applications relating to vaccines made using spontaneous amide bond formation and a shareholder in SpyBiotech. All other authors have declared that no competing interests.
