## [Peer Review File · Nature Communications]

Development of an improved blood-stage malaria vaccine targeting the essential RH5-CyRPA-RIPR invasion complexReviewers' Comments:

Reviewer #1:

Remarks to the Author:

RH5.1/Matrix-M is the most advanced asexual blood-stage malaria vaccine and is currently in a Phase 2b field efficacy trial. To improve the vaccine efficacy over the RH5.1/Matrix-M™, Williams et al. conducted a study to investigate if vaccinating with the RCR complex could enhance the efficacy of the RH5.1 vaccine. They discovered that the RCR-complex exposes surface-inhibitory epitopes on each antigen and that combining different mAb pairs targeting different antigens can function in an additive or synergistic manner. However, they found that immunization with the RCR-complex did not perform better than RH5 alone due to the immuno-dominance of R1PR and the inferior potency of anti-R1PR polyclonal IgG. Further analysis revealed that all growth inhibitory antibody epitopes of R1PR were located in EGF(5-8). Finally, they identified that combining R78C with RH5 resulted in improved growth inhibition activity compared to RH5 alone.

The results of this study are very useful for improving the leading RH5.1 asexual blood-stage malaria vaccine currently in Phase 2b development. All the works are carefully designed and clearly presented, and the manuscript is well-written. I have several comments to improve the manuscript further.

Comments:

1) A Phase 1a study (ClinicalTrials.gov NCT05385471) is currently being conducted based on the promising results of this study. However, it's not entirely clear how to prepare for the formulation of this novel vaccine candidate in Phase 1a. There are two options: either using "RCR-78 mini" mixed with Matrix-M or mixing R78C and RH5.1 with Matrix-M at the same time. It's essential to confirm which formulation was progressed to Phase 1a in the Discussion, along with some reasoning, as there is a slight difference in GIA (not significant though) between the two vaccine formulations (Figure 6).

2) Lines 570-572 "The R78C construct consists of amino acids D817-V897 of R1PR, two GGSGS linkers, the SpyTag peptide, two GGGGS linkers, amino acids D29-E362 of CyRPA and the C-tag at the C-terminus. R58C is the same construct except with amino acids P716-D900 of R1PR." Why did they design this construct? For example, why not C-R78 or C-R58? Why the insertion of "the SpyTag peptide" in between? Please add the reason.

3) Lines 469-470 "Indeed, we could not detect any binding between recombinant R1PR and SEMA7A as previously reported⁴⁸."

The reference of this statement should be #47.

4) Line 656 "PBS containing 1 mM CaCl₂. AP-conjugated anti-human IgG"

Please correct as "PBS containing 1 mM CaCl₂. AP-conjugated anti-rat IgG".

5) Lines 663 & 667-668 "Four 16-week-old New Zealand rabbits" and "using 16-week-old New Zealand white rabbits"

Please add Female or Male. I guess both should be "white" rabbits. Please confirm.

Reviewer #2:

Remarks to the Author:

In this manuscript, the authors have focused on development of multiantigen blood stage malaria vaccine candidate and investigates if RCR complex could induce a more potent growth inhibitory response as compared to its single component such as RH5. PfRH5 or RH5.1 is the most advanced blood-stage vaccine candidate in a Phase 2b clinical trial. RH5 interacts with basigin receptor for merozoite invasion of human RBC's. It has been shown that RH5 induces broadly neutralizing and cross strain protective antibodies that inhibits parasite growth both in-vitro and in various animal models.

Manuscript is well written. Results are presented clearly, and the data is analyzed in detail and systematically using monoclonal antibodies against each antigen involved in RCR complex formation. Authors have clearly defined Type I/II antibodies and the selection process of synergistic antibody pairs. This study shows that both GIA negative (R5.011) or GIA-positive (R5.008) mAbs can synergistically induce growth inhibitory antibodies. However synergistic effect shown through mAbs analysis did not translate in rat study as RCR complex as immunization did not induce synergistic protective antibodies due to immunodominance of RIPR. To overcome this, they have generated as series of RIPR fragments, identified RIPR EGF (7-8) and RIPR EGF (5-8) protein was capable of complete GIA reversal, and created fusion proteins "R78C" and "R58C" that forms complex with RH5. Rabbit immunization shows that R78C+RH5 performed better than R58C+RH5 and RH5 alone with plans to further test R78C+RH5 in Phase 1.a clinical trials. R78C+RH5 performed better than R58C+RH5 in animal studies. Both contains EGF7-8 region. Takashima et. al. also showed that PfRipr_5: C720-D934, a C-terminal EGF-like domains generates potent inhibitory merozoite invasion antibodies. This study further narrows down to EGF7-8 region in RIPR and shows that this region generates the most synergistic and additive antibodies.

Minor Comments:

Line 499 - Mutation N28Q. Is this correct?

Figure 1.D and E - Could add RH5 Type I and Type II as figure title just like the ones in Figures in S1 Supplemental information Page 1- Table S4 and S5 are switched. Table S4 contains details of RIPR truncations produced and Table S5 contains primer information.

C12 and 8A7 CyRPA antibodies shown in Figure S1.B are missing in table S1

Figure S3.B - Last GIA plot does not have figure title.

Figure S3.C - First and last figure does not have figure title.

Response to reviewer comments: NCOMMS-24-10746**Reviewer #1:**

RH5.1/Matrix-M is the most advanced asexual blood-stage malaria vaccine and is currently in a Phase 2b field efficacy trial. To improve the vaccine efficacy over the RH5.1/Matrix-M™, Williams *et al.* conducted a study to investigate if vaccinating with the RCR complex could enhance the efficacy of the RH5.1 vaccine. They discovered that the RCR-complex exposes surface-inhibitory epitopes on each antigen and that combining different mAb pairs targeting different antigens can function in an additive or synergistic manner. However, they found that immunization with the RCR-complex did not perform better than RH5 alone due to the immuno-dominance of RIPR and the inferior potency of anti-RIPR polyclonal IgG. Further analysis revealed that all growth inhibitory antibody epitopes of RIPR were located in EGF(5-8). Finally, they identified that combining R78C with RH5 resulted in improved growth inhibition activity compared to RH5 alone.

The results of this study are very useful for improving the leading RH5.1 asexual blood-stage malaria vaccine currently in Phase 2b development. All the works are carefully designed and clearly presented, and the manuscript is well-written. I have several comments to improve the manuscript further.

We thank the Reviewer for their highly positive comments and summary of our manuscript.

Comments:

1) A Phase 1a study (ClinicalTrials.gov NCT05385471) is currently being conducted based on the promising results of this study. However, it's not entirely clear how to prepare for the formulation of this novel vaccine candidate in Phase 1a. There are two options: either using "RCR-78 mini" mixed with Matrix-M or mixing R78C and RH5.1 with Matrix-M at the same time. It's essential to confirm which formulation was progressed to Phase 1a in the Discussion, along with some reasoning, as there is a slight difference in GIA (not significant though) between the two vaccine formulations (Figure 6).

The Reviewer highlights an important point, and we have now provided clarification on this. For the R78C + RH5.1 groups in the Phase 1a study it was decided to admix the RH5.1, R78C and Matrix-M™ adjuvant at the bedside immediately prior to immunisation rather than produce an "RCR-78 mini" vaccine. This approach was chosen for two reasons: Firstly the lack of significant difference between the "RCR-78 mini" and "RH5.1+R78C mixture" ELISA and GIA data (which we report here and as pointed out by the Reviewer). Secondly, the RH5.1 clinical product used in NCT05385471 was made for a different trial originally under a different programme and was therefore ready to use in the clinic. Funding was available to GMP manufacture a second vaccine candidate (R78C), but given the previously mentioned data, there was no justification for the substantial extra cost required to scale-up and manufacture a "pre-complexed" vaccine. The additional regulatory and cost hurdles of producing such a complexed vaccine outweighed any perceived advantage over admixing the two soluble proteins at the bedside for a proof-of-concept Phase 1a trial.

We have now clarified this in the Discussion and summarized this reasoning.

2) Lines 570-572 "The R78C construct consists of amino acids D817-V897 of RIPR, two GGSGS linkers, the SpyTag peptide, two GGGGS linkers, amino acids D29-E362 of CyRPA and the C-tag at the C-terminus. R58C is the same construct except with amino acids P716-D900 of RIPR." Why did they design this construct? For example, why not C-R78 or C-R58? Why the insertion of "the SpyTag peptide" in between? Please add the reason.

The order of the RIPR EGF(7-8) and CyRPA was initially chosen arbitrarily, however a long linker was put into the construct in an attempt to make all epitopes available on each antigen. We had decided that the order could be reversed if the constructs expressed poorly to try and improve expression however this turned out not to be necessary.

The SpyTag peptide was originally included since, at the initial time of vaccine design, there was significant interest in virus-like particle (VLP)-based vaccines and the SpyTag gave us the option to conjugate the antigen to a VLP-SpyCatcher in the future if deemed necessary. However, since the fusion of RIPR EGF(7-8) to CyRPA was successful in restoring the immunogenicity of the small RIPR EGF(7-8) fragment, and because the RH5.1 construct is not a VLP, it made sense to test all constructs as soluble antigens rather than mix VLP and non-VLP vaccines.

We have now added information to the manuscript to clarify this point.

3) Lines 469-470 "Indeed, we could not detect any binding between recombinant RIPR and SEMA7A as previously reported⁴⁸."

The reference of this statement should be #47.

This has now been corrected.

4) Line 656 "PBS containing 1 mM CaCl. AP-conjugated anti-human IgG". Please correct as "PBS containing 1 mM CaCl₂. AP-conjugated anti-rat IgG".

This has now been corrected. The subsequent product code for the anti-rat IgG has also been corrected.

5) Lines 663 & 667-668 "Four 16-week-old New Zealand rabbits" and "using 16-week-old New Zealand white rabbits". Please add Female or Male. I guess both should be "white" rabbits. Please confirm.

The lines indicated have now been changed to include "female" and "white".

Reviewer #2:

In this manuscript, the authors have focused on development of multiantigen blood stage malaria vaccine candidate and investigates if RCR complex could induce a more potent growth inhibitory response as compared to its single component such as RH5. PfRH5 or RH5.1 is the most advanced blood-stage vaccine candidate in a Phase 2b clinical trial. RH5 interacts with basigin receptor for merozoite invasion of human RBC's. It has been shown that RH5 induces broadly neutralizing and cross strain protective antibodies that inhibits parasite growth both in-vitro and in various animal models.

Manuscript is well written. Results are presented clearly, and the data is analyzed in detail and systematically using monoclonal antibodies against each antigen involved in RCR complex formation. Authors have clearly defined Type I/II antibodies and the selection process of synergistic antibody pairs. This study shows that both GIA negative (R5.011) or GIA-positive (R5.008) mAbs can synergistically induce growth inhibitory antibodies. However synergistic effect shown through mAbs analysis did not translate in rat study as RCR complex as immunization did not induce synergistic protective antibodies due to immunodominance of RIPR. To overcome this, they have generated as series of RIPR fragments, identified RIPR EGF (7-8) and RIPR EGF (5-8) protein was capable of complete GIA reversal, and created fusion proteins "R78C" and "R58C" that forms complex with RH5. Rabbit immunization shows that R78C+RH5 performed better that R58C+RH5 and RH5 alone with plans to further test R78C+RH5 in Phase 1.a clinical trials.

R78C+RH5 performed better that R58C+RH5 in animal studies. Both contains EGF7-8 region. Takashima *et. al.* also showed that PfRipr_5: C720-D934, a C-terminal EGF-like domains generates potent inhibitory merozoite invasion antibodies. This study further narrows down to EGF7-8 region in RIPR and shows that this region generates the most synergistic and additive antibodies.

We thank the Reviewer for their positive comments and summary of our manuscript.

Minor Comments:

Line 499 - Mutation N28Q. Is this correct?

This has been corrected to N228Q.

Figure 1.D and E – Could add RH5 Type I and Type II as figure title just like the ones in Figures in S1

This has been added to Figure 1 D and E as suggested.

Supplemental information Page 1- Table S4 and S5 are switched. Table S4 contains details of RIPR truncations produced and Table S5 contains primer information.

This has been corrected.

C12 and 8A7 CyRPA antibodies shown in Figure S1.B are missing in table S1

These have now been added.

Figure S3.B - Last GIA plot does not have figure title. Figure S3.C - First and last figure does not have figure title.

These have now been added.